# Enhancing Federated Domain Adaptation with Multi-Domain Prototype-Based Federated Fine-Tuning

**Jingyuan Zhang[1], Yiyang Duan[1], Shuaicheng Niu[1], Yang Cao[2], Wei Yang Bryan Lim[1]***

1 College of Computing and Data Science, Nanyang Technological University
2 Department of Computer Science, Institute of Science Tokyo
{jzhang149, yiyang007}@e.ntu.edu.sg, shuaicheng.niu@ntu.edu.sg
cao@c.titech.ac.jp, bryan.limwy@ntu.edu.sg

## Abstract

Federated Domain Adaptation (FDA) is a Federated Learning (FL) scenario where models are trained across multiple clients with unique data domains but a shared category space, without transmitting private data. The primary challenge in FDA is data heterogeneity, which causes significant divergences in gradient updates when using conventional averaging-based aggregation methods, reducing the efficacy of the global model. This further undermines both in-domain and out-of-domain performance (within the same federated system but outside the local client). To address this, we propose a novel framework called **M**ulti-domain **P**rototype-based **F**ederated Fine-**T**uning (MPFT). MPFT fine-tunes a pre-trained model using multi-domain prototypes, i.e., pretrained representations enriched with domain-specific information from category-specific local data. This enables supervised learning on the server to derive a globally optimized adapter that is subsequently distributed to local clients, without the intrusion of data privacy. Empirical results show that MPFT significantly improves both in-domain and out-of-domain accuracy over conventional methods, enhancing knowledge preservation and adaptation in FDA. Notably, MPFT achieves convergence within a single communication round, greatly reducing computation and communication costs. To ensure privacy, MPFT applies differential privacy to protect the prototypes. Additionally, we develop a prototype-based feature space hijacking attack to evaluate robustness, confirming that raw data samples remain unrecoverable even after extensive training epochs. The complete implementation of MPFL is available at `https://ntu-zjy.github.io/DomainFL/`.

## 1 Introduction

Federated Learning (FL) is a privacy-preserving distributed machine learning paradigm designed to protect the data of participating clients (McMahan et al., 2017b). In FL, only models trained on local data are shared between clients and servers, rather than the raw data itself, mitigating the risk of data leaks. Mainstream FL research primarily focuses on optimizing each client's performance within its local data domain (in-domain performance). However, in Federated Domain Adaptation (FDA) scenarios, clients need to perform well on the collective data domains shared by all participants to meet certain business requirements. For instance, consider a consortium of banks training a model to detect fraudulent transactions. Each bank has distinct customer bases and transaction patterns, leading to variations in their data (domains). A good out-of-domain performance is essential for anticipating new risks that a bank may not be exposed to yet. The challenge is to ensure the global model performs well across all banks, achieving good in-domain accuracy (within each bank's typical use cases) and out-of-domain accuracy (generalizing across the federated financial system).

Studies have shown that FL achieves results similar to centralized training only when the datasets among clients are independently and identically distributed (i.i.d) and share similar domain char-

---

*Corresponding author

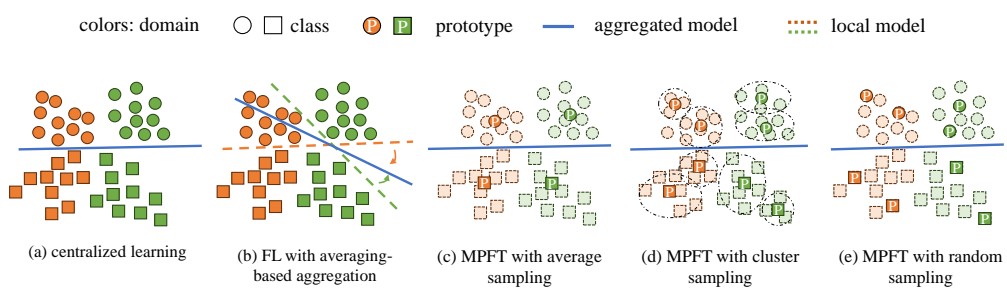

Figure 1: Comparison of MPFT to centralized learning and previous averaging-based FL approaches.

acteristics (Liu et al., 2023; Seol & Kim, 2023; Shaheen et al., 2022). In typical scenarios where all clients' models are trained on local data from the same domains, update directions are aligned, making it feasible to use averaging-based aggregation algorithms such as FedAvg (McMahan et al., 2017b). However, in FDA, in which clients have unique domains but a shared category space, the significant dissimilarities of feature space among local model updates mean that averaging of weights may not yield an optimal global model (Su et al., 2024; Sun et al., 2021).

To address this issue, we propose a novel framework called Multi-domain Prototype-based Federated Fine-Tuning (MPFT). MPFT tackles this issue by having each client generate a specific proportion of data embeddings (i.e., prototypes) to be transmitted to the server to create a prototype training dataset. This allows us to simulate a centralized learning approach without transferring raw data, as prototypes encapsulate sufficient domain-specific features to represent the entire data domain. The server then fine-tunes a global adapter using this comprehensive set of prototypes, with the goal of approximating the performance of centralized learning without relying on conventional averaging approaches to derive a global model. Figure 1 illustrates how our aggregation method compares to centralized learning and traditional averaging-based FL approaches.

Since only a specific proportion of local data embeddings are sampled as prototypes, and MPFT requires only a single round of global communication to converge, our framework incurs significantly lower computation and communication costs compared to other multi-round FL frameworks. MPFT also incorporates a differential privacy mechanism to mitigate the risk that the original data of specific prototypes are exposed during the prototype transmission process. Furthermore, simulations of feature space hijacking attacks on MPFT demonstrate that attackers cannot reconstruct the original data from the uploaded prototypes, even when the pretrained prototype encoder is known.

Our contributions are as follows: a) We propose MPFT, a one-round federated fine-tuning framework with convergence guarantees that outperforms previous methods on multi-domain environments. b) We introduce a novel metric to evaluate the performance of the FDA, specifically assessing out-of-domain and in-domain accuracy to consider the trade-off between knowledge preservation and adaptation. c) We demonstrate empirically that MPFT incurs lower computational and communication overheads as compared to other FL methods while ensuring privacy through differential privacy protection of the prototypes and maintaining robustness against feature space hijacking attacks.

## 2 RELATED WORK

Efforts to enhance performance in FL across heterogeneous datasets (FDA) have been extensive. Regularization methods during training, such as FedProx (Li et al., 2020) and FedDyn (Jin et al., 2023), or knowledge distillation techniques like FedGen (Venkateswaran et al., 2023) and FedNTD (Lee et al., 2022), aim to align local models more closely with the global model. Such approaches help mitigate the divergence among local models but rely on the assumption that the averaged global model is well-suited for the entire data distribution which often does not hold in FDA.

Updating only parts of the model during aggregation is another strategy, exemplified by personalized FL (PFL) (Tan et al., 2022a), which aims to customize local models to enhance in-domain perfor-

mance. Some methods involve using a personalized aggregation base to select specific portions of global information for model aggregation, or updating only parts of the model during the aggregation phase, e.g., APFL (Deng et al., 2020), FedFomo (Zhang et al., 2020), FedAMP (Huang et al., 2021), FedPHP (Li et al., 2021c), APPLE (Luo & Wu, 2022), and FedALA (Zhang et al., 2023a). Additionally, some methods split model layers to segregate global and local components, such as in FedPer (Arivazhagan et al., 2019), LG-FedAvg (Liang et al., 2020), FedRep (Husnoo et al., 2022), FedRoD (Chen & Chao, 2021), FedBABU (Oh et al., 2021), FedCP (Zhang et al., 2023b), FedGH (Yi et al., 2023), and DBE (Zhang et al., 2024). However, a key drawback is these methods still rely on averaging-based aggregation, which results in poor out-of-domain adaptation performance, despite achieving some in-domain improvements.

Some FL methods incorporate prototype learning. For instance, FedProto (Tan et al., 2022b) utilizes averaged local prototypes to train as the global prototype on local clients. FPL (Huang et al., 2023) transmit all the prototypes (data embeddings) to the server during the initial phase for clustering and averaging, which is then sent back to the clients for further optimization. This method introduces significant communication overhead and potential privacy leakage during the first phase. FedNH (Dai et al., 2023) leverages class prototype transmission to address class imbalance across clients, rather than tackling the more complex issue of domain heterogeneity.

Unlike the above methods, we do not assume that the average aggregation of different client models or prototypes can represent the global model or global prototype in scenarios involving heterogeneous client data. MPFT uses prototypes from different clients as basic units that replace data distribution for centralized training on the server, thus obtaining an approximate global model capable of fitting all heterogeneous client domains distribution, as demonstrated in Appendix A.

## 3 PROBLEM STATEMENT

**Scenario assumption.** Consider a scenario involving $N$ clients, where each client possesses a private training dataset $\mathbb{D}_1, \ldots, \mathbb{D}_N$, each drawn from a unique data domain and shared category space[1]. Our goal is to train a model that balances domain knowledge preservation and domain knowledge adaptation. *Domain knowledge preservation* refers to the ability of the model to retain each client's unique domain insights. *Domain knowledge adaptation* is defined as the model's ability to indirectly extract and transfer knowledge from each participating client's domain to others, even if a client $i$ cannot directly train on another client $j$'s data.

**Optimization goal.** Building on this, we define the *Domain Knowledge Preservation* loss $\mathcal{L}^P$ and the *Domain Knowledge Adaptation* loss $\mathcal{L}^A$ as follows:

$$\mathcal{L}^P = \sum_{i=1}^{N} \mathcal{L}_i(\Theta_i^{\mathcal{P}}; \mathbb{D}_i; \Theta^{\mathcal{G}}), \qquad \mathcal{L}^A = \sum_{i=1}^{N} \sum_{j=1, j \neq i}^{N} \mathcal{L}_i(\Theta_i^{\mathcal{P}}; \mathbb{D}_j; \Theta^{\mathcal{G}}), \quad (1)$$

where $\mathcal{L}(\cdot)$ denotes the loss function, $\Theta_i^{\mathcal{P}}$ denotes the local (personalized) model parameters for client $i$, and $\Theta^{\mathcal{G}}$ denotes the global model parameters. In conventional FL, the local models $\Theta_1^{\mathcal{P}}, \ldots, \Theta_N^{\mathcal{P}}$ are periodically synchronized with the global model $\Theta^{\mathcal{G}}$.

The optimization goal is to decrease both the $\mathcal{L}^P$ and $\mathcal{L}^A$. Thus, we formalize the FDA optimization goal as follows:

$$\{\Theta_1^{\mathcal{P}}, \ldots, \Theta_N^{\mathcal{P}}; \Theta^{\mathcal{G}}\} = \arg\min(\alpha_i^1 \mathcal{L}^P + \alpha_i^2 \mathcal{L}^A), \quad (2)$$

where $\alpha_i^1$ and $\alpha_i^2$ are client-defined weight parameters which balance the trade-off between domain knowledge preservation and domain knowledge adaptation, in line with the "no free lunch" theorem.

**Evaluation metrics.** To quantify the effectiveness of the optimization, we propose two metrics: *in-domain accuracy (ind acc)* and *out-of-domain accuracy (ood acc)*. Denoting $\text{ACC}_i^{(j)}$ to be the accuracy for a client $i$ when tested on domain $j$, ind acc and ood acc are defined as follows:

---

[1]These domains may be largely isolated with minimal overlap, but MPFT also generalizes to scenarios where each client may have data from multiple domains, as demonstrated in Section 5.

$$\text{ind acc} = \frac{\sum_{i=1}^{N} \text{ACC}_i^{(i)} n_i}{\sum_{i=1}^{N} n_i}, \qquad \text{ood acc} = \frac{\sum_{i=1}^{N} \sum_{j \neq i} \text{ACC}_i^{(j)} n_j}{\sum_{i=1}^{N} \sum_{j \neq i} n_j}, \qquad (3)$$

where $n_i$ is the number of test samples for client $i$. The *ind acc* measures each client's performance on its own domain data, while the *ood acc* evaluates adaptation performance when tested on data from other domains.

## 4 METHODOLOGY

**Overview.** We introduce Multi-domain Prototype-based Federated Fine-Tuning (MPFT), which consists of three main components: Prototype Generation, Global Adapter Initialization, and Few-shot Local Adaptation, as depicted in Figure 2. During the Prototype Generation phase, we generate *domain-specific prototypes* for each client based on specific sampling methods and ratios. The clients subsequently transmit their local prototypes to the server. In the Global Adapter Initialization phase, we utilize these prototypes to train a global adapter designed to handle the multi-domain distribution of all clients, thereby improving the ood performance of the global adapter across clients. The global adapter is then sent back to the clients for local inference. While the global adapter performs well in ood accuracy, some clients may require better ind performance. In such cases, they can proceed to the Few-shot Local Adaptation phase, where a few-shot dataset is sampled locally to further fine-tune the local adapter. Knowledge distillation is employed to mitigate catastrophic forgetting of global knowledge during this phase.

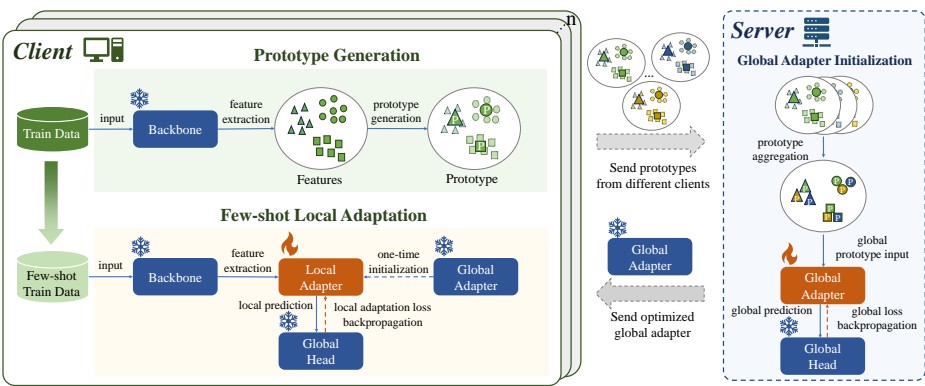

Figure 2: An overview of MPFT.

**Prototype Generation.** A prototype is a compact representation of a specific class feature that is unique to each domain. To synchronize the consistency of the prototype's embedding space across domains, each client utilizes the same pretrained image encoder to generate the prototype set $\{\mathbb{P}^{(1)}, \ldots, \mathbb{P}^{(\mathcal{K})}\}$, where $\mathcal{K}$ represents the total number of classes. Although all clients share the same label space, each label manifests uniquely within its respective feature domain.

To generate these prototypes, different sampling methods can be applied, including *Mean* Sampling, *Cluster* Sampling, and *Random* Sampling, as described in Algorithm 1. The choice of sampling method depends on the desired trade-off between computational efficiency and prototype representational robustness. In *Mean* Sampling, each client $i$ generates a prototype for class $k$ by calculating the mean of the pretrained embeddings for that class. In *Cluster* Sampling, clustering (e.g., k-means) is performed on pretrained embeddings of each class, and a certain number of cluster centers are then selected based on a predefined sampling rate to form the prototype set. In *Random* Sampling, a fixed number of pretrained embeddings of each class are randomly selected according to the sampling rate, and these selected embeddings constitute the prototype set.

Once each client has generated their prototypes, they transmit these to the server. Consequently, the server accumulates $N$ domain-specific representations subset (prototypes subset) for each class $k$, which are collectively represented as $\mathbb{D}^{\mathcal{P}} = \left\{ \bigcup_{i=1}^{N} \left\{ \mathbb{P}_i^{(1)}, \ldots, \mathbb{P}_i^{(\mathcal{K})} \right\} \right\}$.

---

**Algorithm 1** Different Sampling Methods

---

1: **function** MEAN SAMPLING($\mathbb{D}_i$)
2:     **for** class $k$ in $1, \ldots, \mathcal{K}$ **do**
3:         $P_i^{(k)} \leftarrow \frac{1}{|\mathbb{D}_i^{(k)}|} \sum_{(x,y) \in \mathbb{D}_i^{(k)}, y=k} f(\phi; x)$        $\triangleright$ Compute mean embeddings for each class
4:     **end for**
5:     **return** $\mathbb{P}_i \leftarrow \{P_i^{(1)}, \ldots, P_i^{(\mathcal{K})}\}$
6: **end function**

7: **function** CLUSTER SAMPLING($\mathbb{D}_i, r$)
8:     **for** class $k$ in $1, \ldots, \mathcal{K}$ **do**
9:         $C^k \leftarrow \lceil r \times |\mathbb{D}_i^{(k)}| \rceil$           $\triangleright$ Set the number of cluster centers
10:        $\mathbb{P}_i^{(k)} \leftarrow \texttt{Cluster}\left(f(\phi; \mathbb{D}_i^{(k)}), C^k\right)$       $\triangleright$ Perform cluster sampling
11:     **end for**
12:     **return** $\mathbb{P}_i \leftarrow \bigcup_{j=1}^{\mathcal{K}} \mathbb{P}_i^{(j)}$
13: **end function**

14: **function** RANDOM SAMPLING($\mathbb{D}_i, r$)
15:     **for** class $k$ in $1, \ldots, \mathcal{K}$ **do**
16:         $C^k \leftarrow \lceil r \times |\mathbb{D}_i^{(k)}| \rceil$       $\triangleright$ Set the number of randomly selected embeddings
17:        $\mathbb{P}_i^{(k)} \leftarrow \texttt{RandomlySelect}\left(f(\phi; \mathbb{D}_i^{(k)}), C^k\right)$    $\triangleright$ Perform random sampling
18:     **end for**
19:     **return** $\mathbb{P}_i \leftarrow \bigcup_{j=1}^{\mathcal{K}} \mathbb{P}_i^{(j)}$
20: **end function**

---

**Global adapter initialization.** Algorithm 2 outlines the global adapter initialization process. Note that we avoid averaging-based aggregation within each class across different clients, as this would distort the global distribution. Utilizing $\mathbb{D}^{\mathcal{P}}$, we train the global adapter $A^{\mathcal{G}}$ to adapt to the entire system's data distribution with cross-entropy loss $\mathcal{L}$:

$$\{\Theta^{\mathcal{G}}, A^{\mathcal{G}}\} = \arg\min \mathcal{L}(\mathbb{D}^{\mathcal{P}}; \Theta^{\mathcal{G}}; A^{\mathcal{G}}). \tag{4}$$

Upon successful training, the global adapter $A^{\mathcal{G}}$ is sent to the clients, replacing their local adapters.

---

**Algorithm 2** Federated Learning with Global Adapter Initialization

---

**Input:** $N$ clients, $\mathcal{L}$: loss function, $\Theta^0\{f(\phi), g(\phi)\}$: pretrained CLIP model $\Theta^0$ with image encoder $f(\phi)$ and text encoder $g(\phi)$, $A^0$: random initialized adapter, $\eta$: learning rate, $\mathcal{K}$: number of data classes, $\mathbb{D}_i$: client $i$ training data, $(x, y)$: data sample, $\texttt{method}$: sampling method (e.g., $\texttt{mean}$, $\texttt{cluster}$, $\texttt{random}$), $r$: sampling rate.
**Output:** Reasonable global adapter $A^{\mathcal{G}}$
1: Generate linear probe classification head $H$ by labels and pretrained text encoder $g(\phi)$.
2: Server sends $f(\phi)$, $A^0$ and $H$ to all clients to initialize local models.
3: **for** client $i$ in $1, \ldots, N$ in parallel **do**
4:     $\mathbb{P}_i \leftarrow \texttt{method}\,(\mathbb{D}_i, r)$
5: **end for**
6: Clients send the prototype to the server.
7: Server constructs the prototype training dataset $\mathbb{D}^{\mathcal{P}}$ by $\mathbb{D}^{\mathcal{P}} \leftarrow \bigcup_{i=1}^{N} \mathbb{P}_i$.
8: **while** $A^{\mathcal{G}}$ does not converge **do**
9:     Server optimizes $A^{\mathcal{G}}$ by $A^{\mathcal{G}} \leftarrow A^{\mathcal{G}} - \eta \nabla_{A^{\mathcal{G}}} \mathcal{L}(\mathbb{D}^{\mathcal{P}}; \Theta^{\mathcal{G}}; A^{\mathcal{G}})$.
10: **end while**
11: **return** $A^{\mathcal{G}}$

---

**Few-shot local adaptation.** While global adapter $A^G$ performs well in ood accuracy, it may not be sufficient for ind accuracy. To address this, clients can use their local few-shot data $\mathbb{D}_i^{\mathcal{F}}$ to further fine-tune $A^G$, adapting it to their local domain and improving ind accuracy, as shown in Algorithm 3.

---

**Algorithm 3** Few-shot Local Adaptation

---

**Input:** $N$ joining clients, $\mathcal{L}$: loss function, $\Theta^0\{f(\phi), g(\phi)\}$: pretrained CLIP model $\Theta^0$ with image encoder $f(\phi)$ and text encoder $g(\phi)$, $A^G$: global initialized adapter, $\alpha$: local learning rate, $\mathcal{K}$: number of data classes, $\mathcal{F}$: few-shot number, $\mathbb{D}_i$: client $i$ training data, $(x, y)$: data sample.

**Output:** Reasonable local adapters $\{A_1^{\mathcal{L}}, \ldots, A_N^{\mathcal{L}}\}$

1: Server sends $A^G$ to all clients to initialize local adapter $A^{\mathcal{L}}$.
2: **for** client $i$ in $1, \ldots, N$ in parallel **do**
3:     $\mathbb{D}_i^{\mathcal{F}} \leftarrow \left\{ \bigcup_{j=1}^{\mathcal{K}} \{(x_m, y_m) \in \mathbb{D}_i \mid y_m = j, m = 1, \ldots, \mathcal{F}\} \right\}$     ▷ Generate few-shot data
4:     Client $i$ obtains $A_i^{\mathcal{L}}$ by $A_i^{\mathcal{L}} \leftarrow A_i^{\mathcal{L}} - \eta \nabla_{A_i^{\mathcal{L}}} \mathcal{L}(\mathbb{D}_i^{\mathcal{F}}; \Theta^{\mathcal{G}}; A_i^{\mathcal{L}}; A^{\mathcal{G}})$.     ▷ Local adaptation
5: **end for**
6: **return** $\{A_1^{\mathcal{L}}, \ldots, A_N^{\mathcal{L}}\}$

---

To avoid catastrophic forgetting (French, 1999) of global knowledge during the local adaptation process, knowledge distillation (KD) method is employed to regularize the locally trained adapter $A^{\mathcal{L}}$, ensuring it does not deviate too much from the global adapter $A^{\mathcal{G}}$. The loss function for local adaptation is defined as:

$$\mathcal{L} = \mathcal{L}_{\text{CE}} + \beta \mathcal{L}_{\text{KD}}, \tag{5}$$

where $\mathcal{L}_{\text{CE}}$ represents the cross-entropy loss, and $\mathcal{L}_{\text{KD}}$ denotes the KD loss. The hyperparameter $\beta$ balances the influence of the KD loss in the overall objective. In Section 5.4, we compare the effects of different KD weights $\beta$ on balancing ind accuracy and ood accuracy.

**Convergence.** We analyze the convergence of MPFT, with detailed proofs provided in Appendix B.

**Theorem 1 (Convergence of fine-tuning with prototypes)** *For a smooth, non-convex loss function $\mathcal{L}$ with a Lipschitz continuous gradient with constant $L$, the global fine-tuning using prototypes $\mathbb{D}^{\mathcal{P}}$ converges. The sequence of updates for the global adapter $A^{\mathcal{G}}$ achieves a monotonic decrease in the loss function $L(\mathbb{D}^{\mathcal{P}}; \Theta^{\mathcal{G}}, A^{\mathcal{G}})$. Specifically, choosing a learning rate $\eta$ such that $0 < \eta < \frac{2}{L}$ ensures:*

$$\mathcal{L}(\mathbb{D}^{\mathcal{P}}; \Theta^{\mathcal{G}}, A_{t+1}^{\mathcal{G}}) \leq \mathcal{L}(\mathbb{D}^{\mathcal{P}}; \Theta^{\mathcal{G}}, A_t^{\mathcal{G}}) - c\|\nabla_{A^{\mathcal{G}}} \mathcal{L}(\mathbb{D}^{\mathcal{P}}; \Theta^{\mathcal{G}}, A_t^{\mathcal{G}})\|^2 + \frac{\Delta}{N},$$

*where $c = \eta - \frac{L\eta^2}{2}$ is a positive constant, ensuring that the step size is appropriately bounded for convergence. Here, $\Delta$ is the maximum prototype divergence across clients.*

**Corollary 1.1 (Convergence to stationary point and rate)** *As $T$ increases, the average gradient norm decreases, indicating convergence to a stationary point:*

$$\frac{1}{T} \sum_{t=1}^{T} \|\nabla_{A^{\mathcal{G}}} \mathcal{L}(\mathbb{D}^{\mathcal{P}}; \Theta^{\mathcal{G}}, A_t^{\mathcal{G}})\|^2 \leq \frac{\mathcal{L}(\mathbb{D}^{\mathcal{P}}; \Theta^{\mathcal{G}}, A_1^{\mathcal{G}}) - \mathcal{L}(\mathbb{D}^{\mathcal{P}}; \Theta^{\mathcal{G}}, A_T^{\mathcal{G}}) + \frac{T\Delta}{N}}{cT}.$$

*This indicates that as $T$ increases, the right-hand side approaches zero, confirming that the gradient norm diminishes and the updates converge to a stationary point.*

## 5 EXPERIMENT

**Datasets and Models.** To simulate a FDA environment, We use the DomainNet (Peng et al., 2019) and PACS (Li et al., 2017) datasets which are widely used in multi-domain data adaptation. For these datasets, we employ pretrained CLIP models from OpenCLIP (Cherti et al., 2023; Radford et al., 2021; Schuhmann et al., 2022). The image encoder of CLIP for DomainNet is a ViT-B-32 pretrained on the LAION-2B dataset, while for PACS, we use a ConvNeXT-Base pretrained on the LAION-400M dataset as the image encoder.

**Implementation details.** We implement various representative FL algorithms as baselines, including FedAvg (McMahan et al., 2017a), FedProx (Li et al., 2020), Ditto (Li et al., 2021b), MOON (Li et al., 2021a), FedProto (Tan et al., 2022b), and DBE (Zhang et al., 2024), using the PyTorch library

and based on the integrated FL library PFLlib (Zhang et al., 2023c). To simulate the common FL scenario where data resides only on clients, we split the data for each client into a training set (70%), a test set (20%), and a validation set (10%). We evaluate in-domain (ind) and out-of-domain (ood) accuracy following Equation 3. To demonstrate the scalability of our method, we partition the DomainNet dataset, which includes 345 categories, into subsets containing 50, 100, and 150 classes, respectively. To more conveniently compute the convergence time for each FL method and compare the computational and communication costs of them, we introduce an early stopping strategy during training. More details about our experimental setup and baselines can be found in Appendix C.1 and Appendix C.2, with details on the early stopping strategy provided in Appendix C.3.

**Diverse FDA Scenarios.** We conducted experiments and analyses from different perspectives on various potential FDA scenarios. Section 5.1 presents the performance of different FL methods in a basic scenario, where each client is assigned a unique data domain, with minimal overlap between domains. Section 5.2 provides a detailed analysis of the global model's performance on each client, exploring the fairness of different methods in FDA. In Section 5.3, we design a more realistic scenario where each client may hold data from multiple domains, reducing the domain heterogeneity between clients. Appendix F sets up another realistic scenario where multiple clients belong to the same data domain, significantly increasing the number of clients compared to the original setup. In Section 5.4, we investigate the role of local adaptation. Section 5.5 compares the computational and communication costs of different methods. Finally, in Section 5.6 and Appendix G, we explore the differential privacy mechanism in MPFT and its robustness against feature space hijacking attacks.

## 5.1 PERFORMANCE ON MULTI-DOMAIN

We evaluate our method alongside other FL approaches in Table 1, including local training (i.e., each client fine-tunes the pretrained model separately). Empirical results show that local training excels in ind accuracy but performs poorly in ood accuracy. A reason is that local fine-tuning results in catastrophic forgetting (Luo et al., 2023). Personalized FL methods such as FedProto and DBE, which generally maintain a personalized local model for each client, have higher ind accuracy but compromise ood accuracy. In contrast, methods like FedAvg, MOON, and Ditto demonstrate more balanced improvements in both ind and ood accuracies. FedProx, which introduces a regularization term between the global and local models, improves ood accuracy at the expense of ind accuracy. In comparison, our method consistently achieves superior performance in both ood and ind accuracy across all DomainNet subsets. As the subset size of DomainNet increases, we observe variable convergence stability across methods such as FedAvg, FedProx, and Ditto, while DBE demonstrates accelerated convergence. In contrast, our method requires only *one global communication round*, which significantly reduces both computational and communication costs. This benefit is further elaborated in section 5.5.

Table 1: Test accuracy and communication rounds for different FL methods on DomainNet subsets and PACS. The communication rounds are determined using an early stopping strategy, where fewer rounds indicate faster convergence. Additionally, we compare the sensitivity of MPFT to different global convergence thresholds to ensure the robustness of the results across various hyperparameters, as refer to Appendix D.

| | DomainNet: Subset-50 | | | DomainNet: Subset-100 | | | DomainNet: Subset-150 | | | PACS | | |
|---|---|---|---|---|---|---|---|---|---|---|---|---|
| | ood acc | ind acc | rounds | ood acc | ind acc | rounds | ood acc | ind acc | rounds | ood acc | ind acc | rounds |
| local | 0.7361 | 0.8609 | 0 | 0.6554 | 0.8310 | 0 | 0.6003 | 0.8067 | 0 | 0.6547 | 0.9984 | 0 |
| FedAvg (McMahan et al., 2017a) | 0.7902 | 0.7345 | 24 | 0.7628 | 0.6966 | 49 | 0.7263 | 0.6709 | 17 | 0.9725 | 0.9887 | 32 |
| FedProx (Li et al., 2020) | 0.7752 | 0.7178 | 10 | 0.7499 | 0.6827 | 9 | 0.7131 | 0.6569 | 5 | 0.9219 | 0.9659 | 13 |
| Ditto (Li et al., 2021b) | 0.7811 | 0.7624 | 20 | 0.7511 | 0.7182 | 30 | 0.7149 | 0.6904 | 13 | 0.9172 | 0.9930 | 35 |
| MOON (Li et al., 2021a) | 0.7902 | 0.7344 | 28 | 0.7623 | 0.6952 | 16 | 0.7267 | 0.6715 | 31 | 0.9763 | 0.9888 | 42 |
| FedProto (Tan et al., 2022b) | 0.7296 | 0.7696 | 5 | 0.6732 | 0.7385 | 8 | 0.6321 | 0.7073 | 7 | 0.8627 | **0.9963** | 32 |
| DBE (Zhang et al., 2024) | 0.7421 | 0.7622 | 22 | 0.7179 | 0.7233 | 6 | 0.6820 | 0.6956 | 5 | 0.971 | 0.984 | 12 |
| MPFT (Average) | 0.8077 | 0.7813 | 1 | 0.7674 | 0.7399 | 1 | 0.7294 | 0.7099 | 1 | 0.9486 | 0.9703 | 1 |
| MPFT (Cluster, rate=0.1) | 0.7951 | 0.7957 | 1 | 0.7641 | 0.7692 | 1 | 0.7171 | 0.7256 | 1 | 0.9808 | 0.9880 | 1 |
| MPFT (Cluster, rate=0.3) | 0.8204 | **0.8294** | 1 | 0.7766 | 0.7791 | 1 | 0.7430 | 0.7514 | 1 | 0.9841 | 0.9896 | 1 |
| MPFT (Random, rate=0.1) | 0.7953 | 0.7899 | 1 | 0.7566 | 0.7509 | 1 | 0.7194 | 0.7233 | 1 | 0.9829 | 0.9888 | 1 |
| MPFT (Random, rate=0.3) | **0.8236** | **0.8294** | 1 | **0.7803** | **0.7811** | 1 | **0.7469** | **0.7542** | 1 | **0.9887** | 0.9919 | 1 |

## 5.2 DETAILS ABOUT RESULTS ON EACH DOMAIN

To further explain why MPFT achieves better ind and ood accuracy compared to the baselines, we visualize the performance of each domain in Figure 3. Each axis of the radar chart represents a specific data domain (e.g., Real or Painting), with the shape and coverage area of the curves illustrating the global model's performance across these domains. Empirically, the roundness of the curve could reflect the *fairness* of the model across different clients (domains)—the rounder the curve, the more fair the method is in the global distribution, leading to better ood accuracy.

Compared to other FL baselines across different DomainNet subset sizes, MPFT with average sampling method performs exceptionally well in the Quickdraw domain, with a more balanced curve shape. Additionally, MPFT maintains strong performance across other domains relative to the baselines, thereby achieving better overall *fairness*. For more details about the effects of random and cluster sampling compared to average sampling on each domain, please refer to Appendix E.

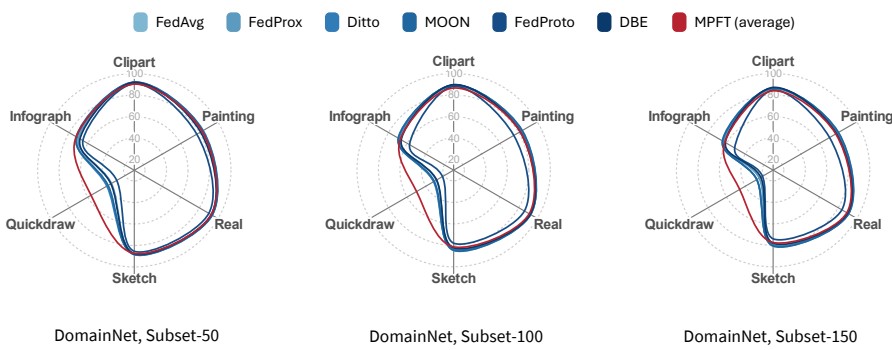

Figure 3: Comparison of different FL methods across various DomainNet subset sizes.

## 5.3 IMPACT OF MULTI-DOMAIN DIFFERENCES ON PERFORMANCE

In real-world scenarios, a client may contain data from multiple domains rather than a single specific domain[2] . We simulate a situation where each client contains $1 - mr$ percent of data from its original domain, mixed with $mr$ percent of data from another domain. Here, $mr$ represents the mixed ratio, indicating the level of domain diversity on the client side. We evaluate our method alongside other FL approaches under this scenario, as shown in Table 2, where DomainNet subset-50 is used. We observe a reduction in the performance advantage of our method compared to others, as the mixed ratio increases. This decline is due to the reduced heterogeneity within the FL system, which diminishes the strengths of our approach. However, it is still evident that our method outperforms most FL algorithms, particularly when using Random and Cluster sampling strategies.

Table 2: Results of mixed multi-domain situation on DomainNet Subset-50.

| | original | | | mixed ratio = 0.3 | | | mixed ratio = 0.4 | | | mixed ratio = 0.5 | | |
|---|---|---|---|---|---|---|---|---|---|---|---|---|
| | ood acc | ind acc | rounds | ood acc | ind acc | rounds | ood acc | ind acc | rounds | ood acc | ind acc | rounds |
| local | 0.7361 | 0.8609 | 0 | 0.7316 | 0.8477 | 0 | 0.7339 | 0.844 | 0 | 0.7408 | 0.8356 | 0 |
| FedAvg (McMahan et al., 2017a) | 0.7902 | 0.7345 | 24 | 0.7975 | 0.7639 | 19 | 0.7996 | 0.7691 | 8 | 0.8090 | 0.7845 | 7 |
| FedProx (Li et al., 2020) | 0.7752 | 0.7178 | 10 | 0.7798 | 0.7386 | 67 | 0.7813 | 0.7448 | 34 | 0.7903 | 0.7588 | 27 |
| Ditto (Li et al., 2021b) | 0.7811 | 0.7624 | 20 | 0.7966 | 0.8027 | 13 | 0.7811 | 0.7808 | 7 | 0.7777 | 0.7757 | 6 |
| MOON (Li et al., 2021a) | 0.7902 | 0.7344 | 28 | 0.7984 | 0.7639 | 13 | 0.8043 | 0.7763 | 8 | 0.8109 | 0.7865 | 8 |
| FedProto (Tan et al., 2022b) | 0.7296 | 0.7696 | 5 | 0.7110 | 0.7395 | 6 | 0.7075 | 0.7421 | 7 | 0.7019 | 0.7293 | 6 |
| DBE (Zhang et al., 2024) | 0.7421 | 0.7622 | 22 | 0.7348 | 0.7371 | 3 | 0.7293 | 0.7349 | 14 | 0.7286 | 0.7331 | 11 |
| MPFT (Average) | 0.8077 | 0.7813 | **1** | 0.7879 | 0.7610 | **1** | 0.7817 | 0.7577 | **1** | 0.7783 | 0.7533 | **1** |
| MPFT (Cluster, rate=0.1) | 0.7951 | 0.7957 | **1** | 0.8213 | 0.8169 | **1** | 0.7887 | 0.7897 | **1** | 0.8007 | 0.7912 | **1** |
| MPFT (Cluster, rate=0.3) | 0.8204 | **0.8294** | **1** | **0.8276** | **0.8253** | **1** | **0.8209** | **0.8142** | **1** | **0.8142** | **0.8071** | **1** |
| MPFT (Random, rate=0.1) | 0.7953 | 0.7899 | **1** | 0.7928 | 0.7856 | **1** | 0.8040 | 0.7958 | **1** | 0.7919 | 0.7801 | **1** |
| MPFT (Random, rate=0.3) | **0.8236** | **0.8294** | **1** | 0.8184 | 0.8141 | **1** | 0.8108 | 0.8065 | **1** | 0.8137 | 0.8050 | **1** |

---

[2]Additionally, it is common for multiple clients to share the same data domain, as refer to Appendix F.

## 5.4 PERFORMANCE WITH LOCAL ADAPTATION

We compare the few-shot performance of local adaptation with different knowledge distillation (KD) weights in Figure 4. As the KD weight increases, there is less out-of-domain knowledge forgetting but worse in-domain knowledge alignment. With an increase in the number of few-shot samples, the ood and ind accuracy show a similar trend. We provide more details about the experiment implementation of KD in local adaptation in Appendix C.4.

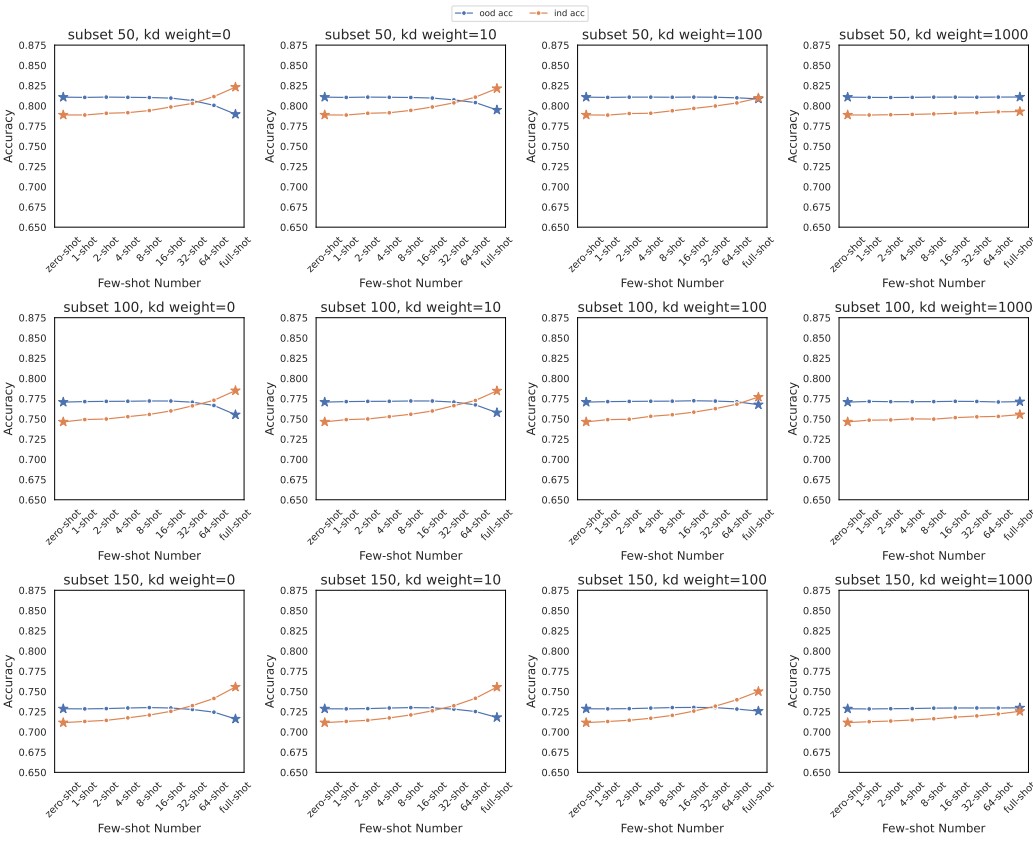

Figure 4: Few-shot performance comparison of local adaptation with different KD weights.

## 5.5 COMPUTATION COST AND COMMUNICATION COST

We further analyze the *computation cost* and *communication cost* in Table 3. *Computation cost* is the total training time of FL, which is related to the number of communication rounds and the computational complexity within each round. Among the methods compared, Ditto is the most time-consuming due to the additional local training epoch it requires. In contrast, the MPFT, which converges in just *one global round*, significantly reduces training time, particularly when the sampling method is set to average. For other sampling methods, such as Random and Cluster, our approach trades a modest increase in training time for substantial improvements in both ind and ood accuracy. *Communication cost* is the number of parameters transmitted, which is theoretically influenced by the number of communication rounds $\mathcal{R}$, the model (adapter) parameters $\sum$, and the prototypes $\prod$. The number of communication rounds $\mathcal{R}$ directly contributes to a linear increase in the overall communication cost. Furthermore, the model adapter parameters $\sum$ and the size of the prototypes $\prod$ determine the communication cost per round in these specific algorithms. In our empirical results, MPFT with average sampling achieved the lowest communication cost across all experiments. However, while random or cluster sampling slightly increases communication overhead, it also significantly improves MPFT's performance (see Table 1).

Table 3: Computation cost and communication cost of FL methods. $\mathcal{R}$ represents the convergence round, $\sum$ represents the number of model (adapter) parameters, $\prod$ represents the size of prototypes, $N$ represents the number of clients. The subsets are derived from the DomainNet dataset.

| | Computation Cost (total training time) | | | | Communication Cost (total parameter transmission) | | | | |
|---|---|---|---|---|---|---|---|---|---|
| | Subset-50 | Subset-100 | Subset-150 | PACS | Subset-50 | Subset-100 | Subset-150 | PACS | Theoretical |
| local | 240.8s | 834.8s | 1575.1s | 311.2s | 0 | 0 | 0 | 0 | 0 |
| FedAvg | 1933.5s | 5812.4s | 3824.6s | 663.9s | 144MB | 294MB | 102MB | 128MB | $\mathcal{R} \times N \times 2\sum$ |
| FedProx | 803.9s | 1253.5s | 1120.1s | 270.0s | 60MB | 54MB | 30MB | 52MB | $\mathcal{R} \times N \times 2\sum$ |
| Ditto | 3095.7s | 8365.8s | 5862.3s | 1455.6s | 120MB | 180MB | 78MB | 140MB | $\mathcal{R} \times N \times 2\sum$ |
| MOON | 2164.2s | 2237.4s | 7027.4s | 872.7s | 168MB | 96MB | 186MB | 168MB | $\mathcal{R} \times N \times 2\sum$ |
| FedProto | 393.1s | 1153.4s | 1608.0s | 683.3s | 5.9MB | 18.8MB | 24.6MB | 3.5MB | $\mathcal{R} \times N \times 2\prod$ |
| DBE | 1693.6s | 837.6s | 1160.3s | 248.8s | 132MB | 36MB | 30MB | 48MB | $\mathcal{R} \times N \times 2\sum$ |
| MPFT (Average) | **1.9s** | **7.3s** | **10.7s** | **0.1s** | **3.6MB** | **4.2MB** | **4.8MB** | **2MB** | $N \times (\prod + \sum)$ |
| MPFT (Cluster, rate=0.1) | 44.7s | 302.4s | 208.0s | 0.6s | 12.5MB | 20.9MB | 30.7MB | 3.1MB | $N \times (\prod + \sum)$ |
| MPFT (Cluster, rate=0.3) | 525.9s | 624.1s | 1344.4s | 2.2s | 30.9MB | 55.8MB | 84.8MB | 5.3MB | $N \times (\prod + \sum)$ |
| MPFT (Random, rate=0.1) | 33.1s | 99.9s | 451.9s | 0.4s | 12.5MB | 20.9MB | 30.7MB | 3.1MB | $N \times (\prod + \sum)$ |
| MPFT (Random, rate=0.3) | 454.5s | 478.9s | 1341.5s | 2.4s | 30.9MB | 55.8MB | 84.8MB | 5.3MB | $N \times (\prod + \sum)$ |

## 5.6 PRIVACY PRESERVATION ANALYSIS

Following DBE (Zhang et al., 2024), we add Gaussian noise $\mathcal{N}$ to client prototypes $p_1, \ldots, p_N$ with a perturbation coefficient $q$ for the noise and a scale parameter $s$ for the noise distribution, the perturbed prototype $\tilde{p}_i$ of client $i$ can be defined as $\tilde{p}_i = p_i + q \cdot \mathcal{N}(0, s^2)$, where $p_i$ is the original prototype of client $i$. The relationship between noise and privacy budget can be found in Appendix H.

Table 4 shows the results of applying this differential privacy method on the DomainNet subset-50, with the sampling method set to average under various noise parameter combinations. This approach effectively mitigates attackers from inferring individual data points even when they possess the pretrained model and most of prototypes. Furthermore, we observe that specific noise configurations can reduce bias across heterogeneous datasets, enhancing the robustness of prototype data. In some cases, this even leads to improved performance compared to models without noise. For instance, the combinations of $(q = 0.5, s = 0.1)$, $(q = 0.1, s = 0.05)$, and $(q = 0.5, s = 0.05)$ exhibit such effects. According to DBE, setting $q = 0.2$ and $s = 0.05$ is sufficient to ensure privacy protection. However, excessively large noise can degrade model performance.

Table 4: Performance of differential privacy with varying noise parameters configuration

| | original | q = 0.1 | | | | s = 0.05 | | | |
|---|---|---|---|---|---|---|---|---|---|
| | | s = 0.1 | s = 0.5 | s = 1 | s = 5 | q = 0.1 | q = 0.2 | q = 0.5 | q = 0.8 |
| ood acc | 0.8077 | 0.8064 | **0.8083** | 0.8065 | 0.7898 | 0.8078 | 0.8064 | **0.8083** | 0.8055 |
| ind acc | 0.7813 | 0.7806 | 0.7806 | 0.7747 | 0.7437 | **0.7824** | 0.7806 | 0.7820 | 0.7782 |

To further evaluate the robustness of MPFT against adversarial attacks, we simulate a feature space hijacking attack (Vepakomma et al., 2021) on MPFT, please refer to Appendix G.

## 6 CONCLUSION AND FUTURE WORK

We propose an adaptive and lightweight FDA framework, MPFT, designed to align a global model with heterogeneous domains by fitting prototypes from different domains. Extensive experiments demonstrate the effectiveness, low cost, and robustness of MPFT. This study may inspire further research in FDA that focuses on generalizing across heterogeneous domain prototypes, rather than relying on model parameter averaging for aggregation.

While MPFT achieves strong performance, it has some limitations. First, the quality of the prototypes is highly dependent on the pretrained model's ability to extract meaningful features. Second, although attackers cannot reconstruct specific raw data from the prototypes, they may still be able to perform membership inference attacks (Shokri et al., 2017) or attribution inference attacks (Fredrikson et al., 2015) by exploiting statistical information contained within the prototypes. Addressing the aforementioned limitations could further enhance the viability and effectiveness of this approach in practical FL applications, making this method a viable alternative of averaging-based FL methods.

ACKNOWLEDGMENTS

This research is supported by the NTU startup grant and the RIE2025 Industry Alignment Fund – Industry Collaboration Projects (IAF-ICP) (Award I2301E0026), administered by A*STAR, as well as supported by Alibaba Group and NTU Singapore through Alibaba-NTU Global e-Sustainability CorpLab (ANGEL). This research / project is supported by A*STAR under its Japan-Singapore Joint Call: JST-A*STAR 2024 (Project ID: R24I6IR139). Any opinions, findings and conclusions or recommendations expressed in this material are those of the author(s) and do not reflect the views of the A*STAR. The authors would also like to thank our lab members Tiantong Wang, Tiantong Wu, Yurong Hao, Mengpu Liu and Wenjie Li for their valuable feedback.

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

## A  VISUALIZATION OF PROTOTYPE

In order to allow readers to intuitively understand the relationship between prototypes and original data, we visualize the data embeddings (all prototypes) and averaged prototypes in two-dimensional coordinates using the t-SNE (Van der Maaten & Hinton, 2008) algorithm, as shown in Figure 5. Different colors represent different domains: for example, blue indicates the "Painting" domain, while orange signifies the "Real" domain. Different markers represent various categories of data samples. The darker markers located within each sample cluster represent the prototypes of the "domain-class". It is evident that each prototype effectively reflects the distribution of its specific domain-class information. However, the mean prototype, represented by the large red star, is distant from each individual prototype. This observation underscores why we do not average the prototypes, as it fails to accurately reflect the overall data distribution, particularly in FDA scenarios.

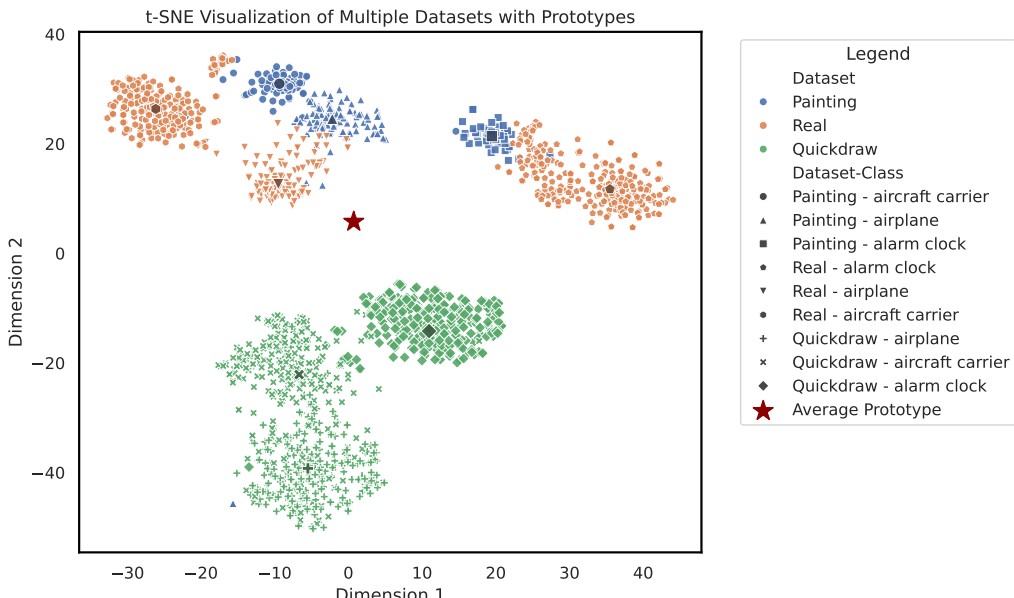

Figure 5: t-SNE visualization of multiple datasets with their corresponding prototypes.

The Universal Approximation Theorem (Hornik et al., 1989) suggests that neural networks act as "universal" data distribution fitters, effectively fitting the distribution of given data samples. However, this also leads to parameter space deviations between different client models in heterogeneous FL. To address these issues, we introduce a more reasonable aggregation method rather than averaging-based aggregation. We treat the same category of data in different domains with significant differences as independent data prototypes. We then use the collection of these data prototypes, which form a more general data distribution that covers the entire FL system, to train a global model that better fits the *global distribution*.

## B CONVERGENCE ANALYSIS

**Setup:** The global loss function for fine-tuning using prototypes is:

$$\mathcal{L}(\mathbb{D}^{\mathcal{P}}; \Theta^{\mathcal{G}}, A^{\mathcal{G}}) = \frac{1}{N} \sum_{i=1}^{N} \mathcal{L}_i(\mathbb{D}^{\mathcal{P}}; \Theta^{\mathcal{G}}, A^{\mathcal{G}}),$$

where $\mathbb{D}^{\mathcal{P}}$ is the set of prototypes aggregated from all clients.

**Assumptions:** The standard assumptions follow those of (Li et al., 2020; Tan et al., 2022b): 1) The loss function $\mathcal{L}$ is non-convex but smooth, 2) The gradient of the loss function is Lipschitz continuous with constant $L$. 3) The divergence between the prototypes of different clients for a given class is bounded. Let $\Delta_i^{(k)}$ be the divergence between client $i$'s prototype for class $k$ and the average prototype across all clients:

$$\Delta_i^{(k)} = \|p_i^{(k)} - \bar{p}^{(k)}\|, \quad \bar{p}^{(k)} = \frac{1}{N} \sum_{i=1}^{N} p_i^{(k)},$$

and assume that $\Delta_i^{(k)} \leq \Delta$ for all $i$ and $k$. Since the prototypes are derived from pretrained image embeddings, this assumption likely holds due to the consistency provided by the pretrained encoder in synchronizing the embedding space across domains.

**Proof:** Given the update rule $A_{t+1}^{\mathcal{G}} = A_t^{\mathcal{G}} - \eta \nabla_{A^{\mathcal{G}}} \mathcal{L}(\mathbb{D}^{\mathcal{P}}; \Theta^{\mathcal{G}}, A_t^{\mathcal{G}})$, for a smooth, non-convex loss function $\mathcal{L}$, the Lipschitz continuity implies:

$$\mathcal{L}(\mathbb{D}^{\mathcal{P}}; \Theta^{\mathcal{G}}, A_{t+1}^{\mathcal{G}}) \leq \mathcal{L}(\mathbb{D}^{\mathcal{P}}; \Theta^{\mathcal{G}}, A_t^{\mathcal{G}}) - \eta \|\nabla_{A^{\mathcal{G}}} \mathcal{L}(\mathbb{D}^{\mathcal{P}}; \Theta^{\mathcal{G}}, A_t^{\mathcal{G}})\|^2 + \frac{L\eta^2}{2} \|\nabla_{A^{\mathcal{G}}} \mathcal{L}(\mathbb{D}^{\mathcal{P}}; \Theta^{\mathcal{G}}, A_t^{\mathcal{G}})\|^2 + \frac{\Delta}{N}.$$

Rearranging the terms, we obtain:

$$\mathcal{L}(\mathbb{D}^{\mathcal{P}}; \Theta^{\mathcal{G}}, A_{t+1}^{\mathcal{G}}) \leq \mathcal{L}(\mathbb{D}^{\mathcal{P}}; \Theta^{\mathcal{G}}, A_t^{\mathcal{G}}) - \left(\eta - \frac{L\eta^2}{2}\right) \|\nabla_{A^{\mathcal{G}}} \mathcal{L}(\mathbb{D}^{\mathcal{P}}; \Theta^{\mathcal{G}}, A_t^{\mathcal{G}})\|^2 + \frac{\Delta}{N}.$$

Choosing $\eta$ such that $0 < \eta < \frac{2}{L}$, we have:

$$\mathcal{L}(\mathbb{D}^{\mathcal{P}}; \Theta^{\mathcal{G}}, A_{t+1}^{\mathcal{G}}) \leq \mathcal{L}(\mathbb{D}^{\mathcal{P}}; \Theta^{\mathcal{G}}, A_t^{\mathcal{G}}) - c \|\nabla_{A^{\mathcal{G}}} \mathcal{L}(\mathbb{D}^{\mathcal{P}}; \Theta^{\mathcal{G}}, A_t^{\mathcal{G}})\|^2 + \frac{\Delta}{N},$$

where $c = \eta - \frac{L\eta^2}{2}$ is a positive constant. This ensures a decrease in the loss function at each step, leading to convergence, with an additional term accounting for prototype divergence.

### B.1 COROLLARY 1: ON CONVERGENCE TO STATIONARY POINT AND CONVERGENCE RATE

To analyze the convergence rate, we sum both sides of the inequality from $t = 1$ to $T$:

$$\sum_{t=1}^{T} \mathcal{L}(\mathbb{D}^{\mathcal{P}}; \Theta^{\mathcal{G}}, A_{t+1}^{\mathcal{G}}) \leq \sum_{t=1}^{T} \mathcal{L}(\mathbb{D}^{\mathcal{P}}; \Theta^{\mathcal{G}}, A_t^{\mathcal{G}}) - c \sum_{t=1}^{T} \|\nabla_{A^{\mathcal{G}}} \mathcal{L}(\mathbb{D}^{\mathcal{P}}; \Theta^{\mathcal{G}}, A_t^{\mathcal{G}})\|^2 + \frac{T\Delta}{N}.$$

This simplifies to:

$$\mathcal{L}(\mathbb{D}^{\mathcal{P}}; \Theta^{\mathcal{G}}, A_1^{\mathcal{G}}) - \mathcal{L}(\mathbb{D}^{\mathcal{P}}; \Theta^{\mathcal{G}}, A_T^{\mathcal{G}}) \geq c \sum_{t=1}^{T} \|\nabla_{A^{\mathcal{G}}} \mathcal{L}(\mathbb{D}^{\mathcal{P}}; \Theta^{\mathcal{G}}, A_t^{\mathcal{G}})\|^2 - \frac{T\Delta}{N}.$$

Rearranging the terms and dividing by $cT$, we obtain the convergence rate:

$$\frac{1}{T} \sum_{t=1}^{T} \|\nabla_{A^{\mathcal{G}}} \mathcal{L}(\mathbb{D}^{\mathcal{P}}; \Theta^{\mathcal{G}}, A_t^{\mathcal{G}})\|^2 \leq \frac{\mathcal{L}(\mathbb{D}^{\mathcal{P}}; \Theta^{\mathcal{G}}, A_1^{\mathcal{G}}) - \mathcal{L}(\mathbb{D}^{\mathcal{P}}; \Theta^{\mathcal{G}}, A_T^{\mathcal{G}}) + \frac{T\Delta}{N}}{cT}.$$

As $T$ increases, the average gradient norm decreases, indicating convergence to a stationary point, adjusted for prototype divergence. Note that the size of the divergence $\Delta$ affects the rate of convergence. A smaller $\Delta$ implies that the prototypes across clients are more similar, leading to faster convergence. Conversely, a larger $\Delta$ suggests greater variability between client data, which may slow down the convergence rate.

## B.2 REMARK: ON ERROR BOUNDS

By incorporating the average prototype divergence, the error bound for client $i$ is:

$$\epsilon_i^{\text{ind}} \leq \epsilon_i^{\text{local}} + \alpha \Delta_i^{\text{avg}},$$

where $\epsilon_i^{\text{local}}$ is the local model error without federation, $\alpha$ is a constant that measures the sensitivity of the error to the prototype divergence, and $\Delta_i^{\text{avg}} = \frac{1}{K} \sum_{k=1}^{K} \Delta_i^{(k)}$ is the average prototype divergence for client $i$. Thus, the in-domain error bound can be written as:

$$\epsilon^{\text{ind}} \leq \frac{\sum_{i=1}^{N} \left( \epsilon_i^{\text{local}} + \alpha \Delta_i^{\text{avg}} \right) n_i}{\sum_{i=1}^{N} n_i}.$$

For out-of-domain accuracy (ood acc), we are interested in how well the global model generalizes across different client domains. The maximum divergence of prototypes captures the worst-case divergence between any two domains.

Given the metric:

$$\text{ood acc} = \frac{\sum_{i=1}^{N} \sum_{j \neq i} \text{ACC}_i^{(j)} n_j}{\sum_{i=1}^{N} \sum_{j \neq i} n_j},$$

By incorporating the maximum prototype divergence, the error bound for client $i$ on domain $j$ is:

$$\epsilon_{ij}^{\text{ood}} \leq \epsilon_i^{\text{local}} + \beta \Delta_{ij}^{\text{max}},$$

where $\epsilon_i^{\text{local}}$ is the local model error without federation, $\beta$ is a constant that measures the sensitivity of the error to the prototype divergence, and $\Delta_{ij}^{\text{max}} = \max_k \|p_i^{(k)} - p_j^{(k)}\|$.

Thus, the overall out-of-domain error bound is:

$$\epsilon^{\text{ood}} \leq \frac{\sum_{i=1}^{N} \sum_{j \neq i} \left( \epsilon_{ij}^{\text{local}} + \beta \Delta_{ij}^{\text{max}} \right) n_j}{\sum_{i=1}^{N} \sum_{j \neq i} n_j}.$$

We note that the in- and out-of-domain error bounds are directly influenced by average prototype divergence and maximum prototype divergence (worst-case scenario), respectively.

- **Impact of higher sampling rate:** A higher sampling rate generally leads to better accuracy because more prototypes are generated per class, providing a richer representation of the feature distribution across domains. This helps the global model better generalize to out-of-domain data by capturing a diverse set of variations.

- **Sampling methods:** Cluster sampling is particularly effective for OOD accuracy because it captures the underlying structure of the data distribution within each class by selecting multiple representative prototypes (e.g., cluster centers). Mean sampling, while computationally efficient, oversimplifies the data distribution by averaging all data points, leading to a loss of critical information needed for robust adaptation. Random sampling performs almost as well as cluster sampling in some scenarios. This may be due to the fact that random sampling, by chance, captures sufficient variations within each class, providing a diverse enough representation to improve OOD accuracy. However, it may not be as reliable as cluster sampling because it lacks systematic selection of prototypes and could miss important subgroups within the class distribution.

## B.3 CONVERGENCE ANALYSIS WITH DIFFERENTIAL PRIVACY NOISE

The convergence analysis changes slightly when differential privacy (DP) noise is added. The key difference is the adjustment in the divergence term:

The divergence between client prototypes for class $k$ is given by:

$$\Delta_i^{(k)} = \|p_i^{(k)} - \bar{p}^{(k)}\|, \quad \bar{p}^{(k)} = \frac{1}{N}\sum_{i=1}^{N} p_i^{(k)}.$$

Suppose Gaussian noise $\beta(p_i^{(k)})$, with an upper bound of $\beta$, is added to each prototype $p_i^{(k)}$. Since the Gaussian noise has a mean of zero, by the law of large numbers, we can assume that $\bar{p}^{(k)}$ remains approximately unchanged when $N$ is large. The new divergence term becomes:

$$\tilde{\Delta}_i^{(k)} \leq \Delta_i^{(k)} + \beta,$$

leading to an updated upper bound:

$$\Delta \leq \Delta + \beta.$$

The new convergence inequality, adapted from Theorem 1, is:

$$\mathcal{L}(\mathbb{D}^{\mathcal{P}};\Theta^{\mathcal{G}}, A_{t+1}^{\mathcal{G}}) \leq \mathcal{L}(\mathbb{D}^{\mathcal{P}};\Theta^{\mathcal{G}}, A_t^{\mathcal{G}}) - c\|\nabla_{A^{\mathcal{G}}}\mathcal{L}(\mathbb{D}^{\mathcal{P}};\Theta^{\mathcal{G}}, A_t^{\mathcal{G}})\|^2 + \frac{\Delta + \beta}{N}.$$

This indicates that the constant term of convergence increases linearly with the noise level. However, since the coefficient of $\beta$ is $\frac{1}{N}$, the impact of DP noise becomes negligible as $N$ grows larger, meaning that the addition of DP noise does not significantly hinder convergence in theory.

## C  MORE DETAILS ABOUT EXPERIMENT IMPLEMENTATION

### C.1  BASIC SETUP

**Global classification head generation.**    We generate the global classification head by utilizing the manually designed prompts and class names of the dataset, calculated by the pretrained text encoder in the CLIP model used in our experiments. The global classification head is obtained by averaging the 'prompt+label' embeddings from all different domains. Following common fine-tune settings, we only train the adapter, while freezing the entire image encoder and global classification head.

**Training.**    Our simulations are conducted on a Google® Compute Platform (GCP) equipped with 47 Intel®Xeon® CPUs and 4 NVIDIA® L4 GPUs. For global adapter training, we employ cross-entropy loss with an AdamW optimizer, setting the learning rate to 0.001. We set the maximum global rounds to 200 and implement an early stopping strategy to evaluate the convergence rounds. It is important to note that our method achieves convergence in just *one* global round, rendering the early stopping strategy primarily applicable to other FL methods. For simplicity, we assume that all clients can participate in every communication round in all experiments.

### C.2  DETAILS ABOUT IMPLEMENTATION OF BASELINES

For the baseline models used in the experiments, we identified the best parameters for our dataset within the recommended parameter ranges provided in their original texts. For FedAvg, we followed the settings in the original article and used the dataset sizes of different clients as the basis for the weighted average. In FedProx, we set the regularization coefficient to 5, which is lower than the usual settings of 10, 100, or 1000. This adjustment was made because a higher regularization coefficient made it difficult for the model to converge to the global equilibrium point due to data heterogeneity. In Ditto, we used a local round number of 1 and set the regularization term to 2. For MOON, we set the coefficient $\mu$ to 0.001 and the temperature coefficient $\tau$ to 1, both within the recommended ranges of the original text. For FedProto, we used 50 as the regularization coefficient. In DBE, we adopted 0.01 as the momentum coefficient and 1 as the regular Xiang coefficient, both within the recommended ranges of the paper. For all baselines, we use the same set of hyperparameters, as shown in Table 5.

To ensure fairness in the comparison, all baseline methods, including FedAvg and FedProx, were trained using the same pre-trained feature extractor as MPFT. Furthermore, the training setup for all

Table 5: The hyperparameters used in all baselines.

| | |
|---|---|
| Optimizer | AdamW |
| Learning rate | 0.001 |
| Batch size | 32 |
| Gradient clip | 1 |
| Local epoch | 1 |
| Maximum global rounds | 200 |
| Warm-up global rounds | 10 |
| Patience global rounds | 10 |

baselines was aligned with MPFT's training paradigm: the pre-trained feature extractor and the global head were frozen, and only the adapter was trained. For consistency, operations in other baselines (e.g., average aggregation in FedAvg) were applied specifically to the adapter parameters instead of the entire model parameters. This adaptation ensures a fair comparison of the performance of MPFT and the baselines under the same conditions.

## C.3 DETAILS ABOUT EARLY STOPPING STRATEGY

Each client completes one local epoch per global round. We set the total number of global rounds to 200 and implement an early stopping strategy to evaluate the convergence rounds of each algorithm. The criterion for early stopping is based on validation loss; specifically, we select the results from the round that achieves the best validation loss as the final outcome. The patience parameter is set to 10 rounds, meaning that if the validation loss does not decrease below the best recorded loss within a span of 10 consecutive rounds, the training process is terminated. By implementing the early stopping strategy, we can more easily test the convergence round of each method and use this strategy to find the round with the best result.

## C.4 DETAILS ABOUT KNOWLEDGE DISTILLATION IN LOCAL ADAPTATION

We use the most basic form of knowledge distillation strategy in our framework (Hinton et al., 2015), which is response-based knowledge distillation:

$$A_i^{\mathcal{L}} = \arg\min \mathcal{L}_{\text{KD}}(\mathbb{D}_i^{\mathcal{F}}; \Theta^{\mathcal{G}}; A_i^{\mathcal{L}}; A^{\mathcal{G}}). \tag{6}$$

Here, $A_i^{\mathcal{L}}$ represents the local adapter for client $i$, $\mathbb{D}_i^{\mathcal{F}}$ is the local dataset, $\Theta^{\mathcal{G}}$ are the global model parameters, and $A^{\mathcal{G}}$ represents the global adapter. Then, we have:

$$I^{\mathcal{G}} = \{A^{\mathcal{G}}(f(\phi; x_1)), \ldots, A^{\mathcal{G}}(f(\phi; x_n))\}, \quad \{(x_1, y_1), \ldots, (x_n, y_n)\} \in \mathbb{D}_i^{\mathcal{F}}. \tag{7}$$

Here, $I^{\mathcal{G}}$ denotes the set of outputs from the global adapter for the local dataset $\mathbb{D}_i^{\mathcal{F}}$.

$$I_i^{\mathcal{L}} = \{A_i^{\mathcal{L}}(f(\phi; x_1)), \ldots, A_i^{\mathcal{L}}(f(\phi; x_n))\}, \quad \{(x_1, y_1), \ldots, (x_n, y_n)\} \in \mathbb{D}_i^{\mathcal{F}}. \tag{8}$$

Similarly, $I_i^{\mathcal{L}}$ denotes the set of outputs from the local adapter $A_i^{\mathcal{L}}$ for the local dataset $\mathbb{D}_i^{\mathcal{F}}$.

$$\mathcal{L}_{\text{KD}} = \text{KL}(I^{\mathcal{G}} \parallel I_i^{\mathcal{L}}) \tag{9}$$

The knowledge distillation loss $\mathcal{L}_{\text{KD}}$ is computed as the Kullback-Leibler (KL) divergence between the outputs of the global adapter and the local adapter.

$$\text{KL}(p \parallel q) = \sum_i p_i \log\left(\frac{p_i}{q_i}\right) \tag{10}$$

Here, $p$ and $q$ represent the probability distributions output by the global and local adapters, respectively, and $\text{KL}(p \parallel q)$ denotes the KL divergence.

# D    SENSITIVITY ANALYSIS OF GLOBAL CONVERGENCE THRESHOLD

During the global adapter initialization phase, we set a threshold to ensure the model stops training the prototypes when the variance of loss over multiple rounds decreases to a low value, indicating that the global adapter has converged. To test the sensitivity of the experimental results to the threshold, we test different thresholds for the global adapter initialization process, specifically 0.1, 0.01, 0.001, and 0.0001. Figure 6 shows the impact of these thresholds[3]. As the threshold decreases, the out-of-distribution (ood) and in-distribution (ind) performance initially increase and then decrease. In contrast, the convergence epochs and training time consistently increase. This trend is intuitive because a lower threshold requires more rounds for the model to converge. Overall, we recommend using a threshold of 0.01 or 0.001 to minimize training time and reduce the risk of overfitting.

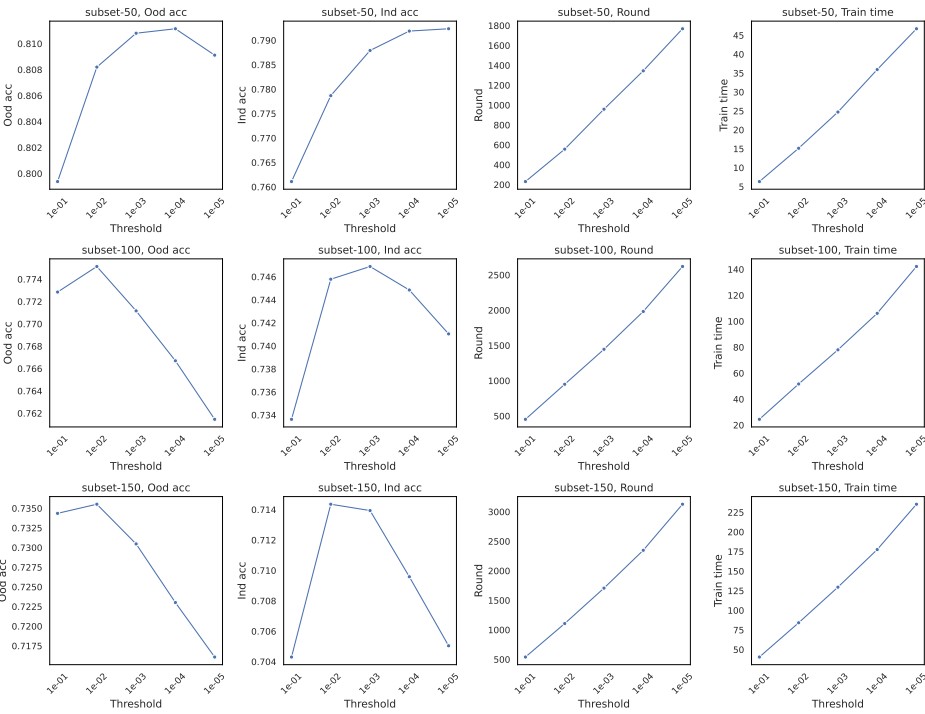

Figure 6: Impact of different convergence thresholds on global adapter initialization.

# E    MORE DETAILS ABOUT RESULTS ON EACH DOMAIN

Figure 7 and Figure 8 further illustrate the effects of random sampling and cluster sampling compared to average sampling within the MPFT framework across different sizes of the DomainNet subset. They reveal similar trends: random sampling and cluster sampling achieve more balanced performance in the Quickdraw domain, with curves approaching circular shapes and covering larger areas. This suggests that these improved sampling methods enhance the model's ability to handle balance and diversity across various data domains.

We also observe that as the sampling rate increases in the random or cluster sampling methods, the model's performance in the Quickdraw domain improves, leading to more globally optimized results. However, the increase in sampling results in more training convergence time consumption and higher data transmission between the server and clients, which raises both computation and communication costs, as shown in Table 3. This trade-off needs to be considered in practical applications.

---

[3]Note that the rounds in the figure represent the convergence epochs for the global prototype training in the server, not the communication rounds (global rounds).

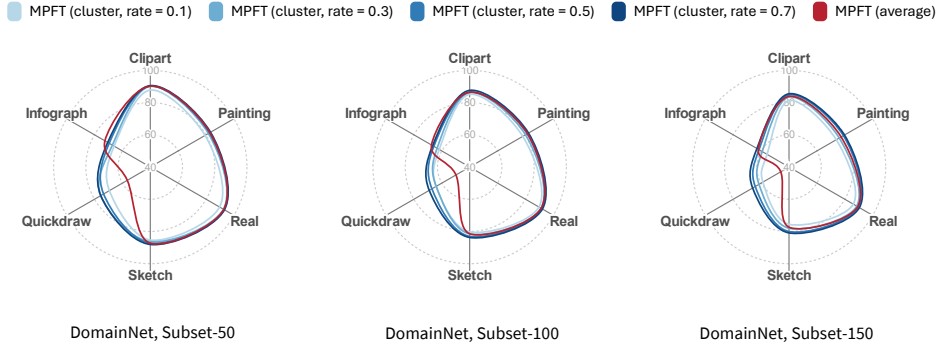

Figure 7: Comparison of cluster and average sampling in MPFT framework.

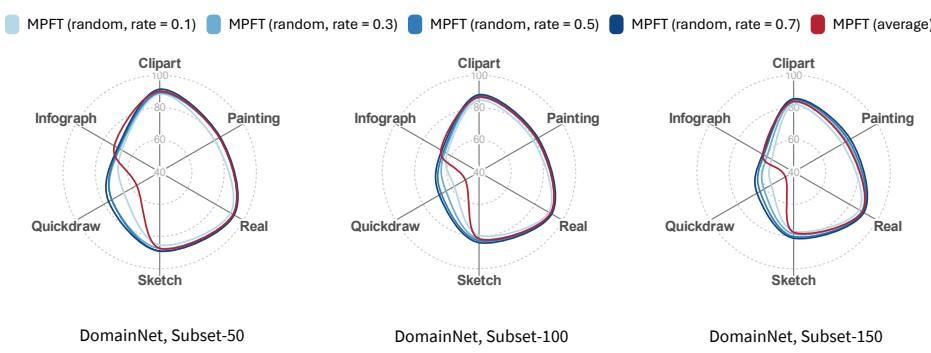

Figure 8: Comparison of random and average sampling in MPFT framework.

## F  RESULTS OF MULTIPLE CLIENTS WITH SAME DOMAIN

In this section, we present the results of experiments where multiple clients belong to the same data domain, a scenario commonly encountered in FL. Specifically, we partition the data from the same domain into multiple subsets, each maintaining the same class labels, and distribute these subsets across multiple clients. This setup results in a far greater number of clients than in the experiments discussed in Section 5.1. As the number of clients increases, the data distribution among clients becomes more similar, potentially reducing domain heterogeneity.

In particular, we compare the results for scenarios with 6, 18, 24, and 30 clients on the DomainNet Subset-50 dataset. Table 6 shows the ind acc, ood acc and the number of communication rounds required for different FL methods.

As the number of clients increases, we observe that methods such as FedAvg, FedProx, and MOON experience a decline in ood accuracy. This could be due to the reduced heterogeneity among clients, leading to less domain-specific knowledge being aggregated into the global model. Conversely, Ditto, FedProto and DBE show a reduction in ind accuracy, because these more personalized methods are less effective at preserving in-domain knowledge when client heterogeneity decreases.

In contrast, MPFT manages to maintain both ind acc and ood acc, similar to the original experimental setup, even as the number of clients increases. Notably, in certain cases, MPFT even improves accuracy. For example, with cluster sampling at a rate of 0.1, MPFT achieves a significant performance boost with 18 clients compared to the original experiment. This suggests that MPFT's adaptive aggregation mechanism is robust to changes in client numbers and data distribution, making it more scalable and effective in scenarios with multiple clients from the same domain.

Table 6: Results of multi clients with same domain on DomainNet Subset-50 dataset.

| | 6 clients (original) | | | 18 clients | | | 24 clients | | | 30 clients | | |
|---|---|---|---|---|---|---|---|---|---|---|---|---|
| | ood acc | ind acc | rounds | ood acc | ind acc | rounds | ood acc | ind acc | rounds | ood acc | ind acc | rounds |
| local | 0.7361 | 0.8609 | 0 | 0.7330 | 0.8557 | 0 | 0.7437 | 0.8479 | 0 | 0.7434 | 0.8451 | 0 |
| FedAvg | 0.7902 | 0.7345 | 24 | 0.7708 | 0.7197 | 25 | 0.7693 | 0.7186 | 24 | 0.7685 | 0.7192 | 25 |
| FedProx | 0.7752 | 0.7178 | 10 | 0.7685 | 0.7165 | 0 | 0.7675 | 0.7163 | 0 | 0.7667 | 0.7172 | 1 |
| Ditto | 0.7811 | 0.7624 | 20 | 0.7643 | 0.7386 | 15 | 0.7639 | 0.7397 | 37 | 0.7619 | 0.7357 | 14 |
| MOON | 0.7902 | 0.7344 | 28 | 0.7709 | 0.7197 | 27 | 0.7691 | 0.7186 | 20 | 0.7682 | 0.7188 | 6 |
| FedProto | 0.7296 | 0.7696 | 5 | 0.7299 | 0.7530 | 9 | 0.7305 | 0.7476 | 9 | 0.7308 | 0.7562 | 11 |
| DBE | 0.7421 | 0.7622 | 22 | 0.7504 | 0.7602 | 8 | 0.7621 | 0.7511 | 4 | 0.7449 | 0.7580 | 24 |
| MPFT (Average) | 0.8077 | 0.7813 | **1** | 0.8062 | 0.7833 | **1** | 0.8032 | 0.7820 | **1** | 0.8032 | 0.7839 | **1** |
| MPFT (Cluster, rate=0.1) | 0.7951 | 0.7957 | **1** | 0.8091 | 0.8128 | **1** | 0.8002 | 0.8011 | **1** | 0.8217 | 0.8232 | **1** |
| MPFT (Cluster, rate=0.3) | 0.8204 | **0.8294** | **1** | 0.8139 | 0.8188 | **1** | 0.8103 | **0.8157** | **1** | 0.8126 | 0.8183 | **1** |
| MPFT (Random, rate=0.1) | 0.7953 | 0.7899 | **1** | 0.7911 | 0.7880 | **1** | 0.7909 | 0.7838 | **1** | 0.7975 | 0.7971 | **1** |
| MPFT (Random, rate=0.3) | **0.8236** | **0.8294** | **1** | **0.8218** | **0.8267** | **1** | **0.8119** | 0.8138 | **1** | **0.8219** | **0.8257** | **1** |

## G  FEATURE SPACE HIJACKING ATTACK

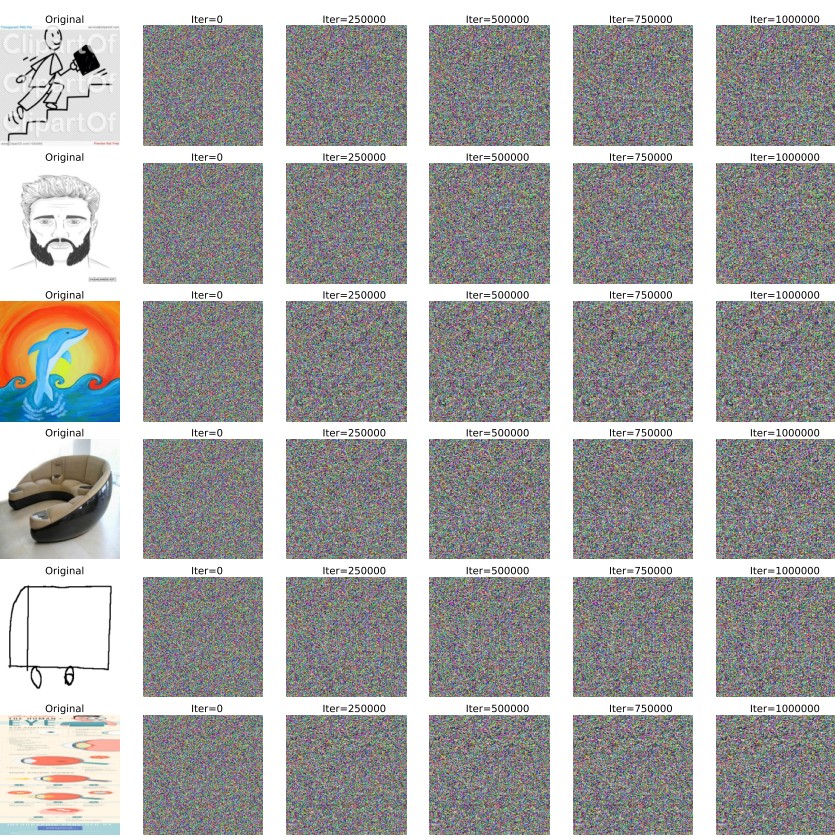

Figure 9: Results of the feature space hijacking attack on our model. Each prototype was constructed from a *single image*. Even after one million iterations of training, the original image could not be recovered, demonstrating the security of using prototypes.

In the MPFT framework, the communication of local clients' prototypes and a trained global adapter may present potential security vulnerabilities. We hypothesize an attack vector utilizing the architecture of our approach, to design a feature space hijacking attack. The attacker could leverage the pretrained model's image encoder $f(\phi)$ and the uploaded prototype $p^{(k)}$ to attempt restoration of the original training data $x$:

1. An estimated input $x^*$ is constructed and processed through the pretrained image encoder $f(\phi)$ to obtain an estimated prototype $p^*$.

2. The mean squared error (MSE) loss is used to iteratively refine $x^*$, aiming to minimize the discrepancy between $p^*$ and the actual prototype $p$, thereby approximating $p_i^{(k)}$.

To demonstrate the resistance of our method to feature space hijacking attacks, we randomly selected a picture from each client to form a prototype and attempted to simulate an attacker trying to restore the picture. One picture is chosen as the prototype because if a prototype attack composed of a single picture cannot be restored, a prototype composed of multiple average representations of the same category will be even more challenging for an attacker to restore and exploit.

Figure 9 illustrates the process of restoring a single image by multiple clients. Intuitively, we observe that even after one million gradient descent iterations, the attacker still cannot restore the salient features of the original image. It is important to note that on our device, it takes nearly 8 hours to complete such iterative training for each image, imposing a significant time cost on attackers attempting large-scale attacks.

# H  RELATIONSHIP BETWEEN NOISE AND PRIVACY BUDGET

We perform differential privacy (DP) analysis in the average prototype sampling stage:

**Proposition 1 (Post-Processing)** *Let $f : \mathbb{N}^{|\mathcal{X}|} \to R$ be a randomized algorithm that is $(\varepsilon, \delta)$-differentially private. Let $g : R \to R'$ be an arbitrary randomized mapping. Then $g \circ f : \mathbb{N}^{|\mathcal{X}|} \to R'$ is $(\varepsilon, \delta)$-differentially private.*

In MPFT, we can regard sampling as $f$ and learning from the prototype as $g$, which output is the learned model. Then the whole learning process is $(\varepsilon, \delta)$-differentially private provided that the sampling function $f$ is $(\varepsilon, \delta)$-differentially private.

**Definition 1 (Gaussian Mechanism)** *Let $f : \mathbb{N}^{|\mathcal{X}|} \to \mathbb{R}^d$ be an arbitrary $d$-dimensional function, and define its $\ell_2$ sensitivity to be:*

$$\Delta_2 f = \max_{adjacent\ x,y} \|f(x) - f(y)\|_2.$$

*The Gaussian Mechanism with parameter $\sigma$ adds noise scaled to $\mathcal{N}(0, \sigma^2)$ to each of the $d$ components of the output.*

**Theorem 2 (Relationship between Gaussian Mechanism and privacy budget)** *Let $\varepsilon \in (0, 1)$ be arbitrary. For $c^2 > 2\ln(1.25/\delta)$, the Gaussian Mechanism with parameter*

$$\sigma \geq c \frac{\Delta_2 f}{\varepsilon}$$

*is $(\varepsilon, \delta)$-differentially private.*

For average sampling, the sampling procedure for class $i$ involves the data embeddings $D_i = \{d_k^{(i)}\}_{k=1}^{n_i}$, where $n_i$ is the number of data points in class $i$. For cluster sampling, simply replace $D_i$ by data embeddings inside the cluster, with other analysis unchanged.

The average sampling function $f$ is

$$f(D_i) = \frac{\sum_{k=1}^{n_i} d_k^{(i)}}{n_i}.$$

We define $f_i$ as the restriction of $f$ taking only data points in class $i$ as input. Two adjacent datasets differ by exactly one data point. For two adjacent datasets $D_p, D_q$, suppose they differ at datapoints $d_p^{(i)} \in D_p, d_q^{(i)} \in D_q$, then:

$$\|f(D_p) - f(D_q)\|_2 = \left\| \frac{\sum_{d_k^{(i)} \in D_p} d_k^{(i)}}{n_i} - \frac{\sum_{d_k^{(i)} \in D_q} d_k^{(i)}}{n_i} \right\|_2 = \frac{\left\| d_p^{(i)} - d_q^{(i)} \right\|_2}{n_i},$$

$$\Delta_2 f_i = \max_{p,q \in \{1,\ldots,n_i\}} \frac{\left\| d_p^{(i)} - d_q^{(i)} \right\|_2}{n_i}.$$

For $(\delta_i, \varepsilon_i)$-differentially private guarantee:

$$\sigma \geq \frac{\sqrt{2\ln(1.25/\delta_i)} \cdot \Delta_2 f_i}{\varepsilon_i}.$$

A strong differentially private guarantee is achieved when $\varepsilon_i < 1$, $\delta_i \leq \frac{1}{n_i}$ according to the theorem in (Dwork & Roth, 2014). In federated learning settings such as (Wei et al., 2020), a medium differentially private guarantee is achieved when $\varepsilon_i$ is around 10.

We calculate our differentially private guarantee as

$$\varepsilon_i = \frac{\sqrt{2\ln(1.25 n_i)} \cdot \Delta_2 f_i}{\sigma}.$$

For our differential privacy experiment in Section 5.6, the $\sigma$ value used is the multiplication $qs$. Table 7 shows empirical results of average privacy budget $\bar{\varepsilon}$ with different $\sigma$ values, where we take $\delta_i = \frac{1}{n_i}$:

Table 7: Average privacy budget $\bar{\varepsilon}$ for different $\sigma$ values.

|  | $\sigma = 0.01$ | $\sigma = 0.05$ | $\sigma = 0.1$ | $\sigma = 0.5$ |
|---|---|---|---|---|
| DomainNet, Subset=50 | 54.76 | 10.95 | 5.48 | 1.10 |
| DomainNet, Subset=100 | 65.48 | 13.10 | 6.55 | 1.31 |
| DomainNet, Subset=150 | 60.73 | 12.15 | 6.07 | 1.21 |

The table demonstrates that, as the noise scale $\sigma$ increases, the average privacy budget $\bar{\varepsilon}$ decreases, indicating stronger privacy protection. When $\sigma$ is above 0.05, the privacy budget is sufficient to provide robust privacy guarantees. Furthermore, as shown in Table 4, the performance of MPFT experiences minimal decline when $\sigma$ is in the range of 0.05–0.1. These results, both theoretical and empirical, demonstrate that MPFT can effectively utilize differential privacy mechanisms to protect prototypes without significant performance loss.

