# OpenReview forum: "Enhancing Federated Domain Adaptation with Multi-Domain Prototype-Based Federated Fine-Tuning"
_ICLR.cc/2025/Conference — ICLR 2025 Poster_

### Official Review · Reviewer_ckUs · 2024-10-28

**Soundness:** 3
**Presentation:** 3
**Contribution:** 3
**Rating:** 8
**Confidence:** 4

**Summary:**

This paper introduces Multi-domain Prototype-based Federated Fine-Tuning (MPFT), a framework designed to address data heterogeneity and data privacy in federated learning (FL) by using prototype-based training rather than traditional averaging methods. In MPFT, each client generates a set of representative data embeddings (prototypes) that capture essential domain-specific characteristics without transferring raw data. These prototypes are then aggregated at the server, allowing for a simulated centralized learning approach and enabling fine-tuning of a global adapter. This method aims to achieve performance on par with centralized learning while solving the challenges incurred by aggregating client models.
The efficiency of MPFT lies in that it requires only a single round of global communication, significantly reducing computational and communication costs compared to multi-round FL methods. Furthermore, by selectively sampling prototypes, the framework limits data transfer volumes. To ensure privacy, MPFT integrates differential privacy mechanisms, mitigating risks of data exposure and rendering prototype-based data reconstruction ineffective—even when the prototype encoder is known to potential attackers.

**Strengths:**

+ They propose an interesting idea -- MPFT as a one-round federated fine-tuning approach for multi-domain environments, demonstrating its performance improved over previous methods.
+ A new metric is introduced to evaluate model adaptability, assessing both out-of-domain and in-domain accuracy to balance knowledge retention and domain adaptation.
+ The writing and logic are clear and easy to follow. The problem formulation is clear and friendly to readers.

**Weaknesses:**

- The proof of convergence in Section B lacks the differential-privacy-related analysis. A supplemental analysis of "MPFT with DP" should be added for theoretical completeness. Intuitively, the addition of differential privacy introduces a bounded randomness during the convergence process.

- The selected experimental counterparts for comparison are mostly in 2017-2022, which are relatively not new. In Section 2, the authors think the "averaging-based aggregation results in poor out-of-domain adaptation performance." It would be better to show the performance of out-of-domain adaptation from the recent advances atop model/parameter aggregation. Some recent works (with or without aggregation) relevant to data heterogeneity in federated learning are expected to compare, such as,

Fedcp: Separating feature information for personalized federated learning via conditional policy. In KDD 2023.
FedFed: Feature Distillation against Data Heterogeneity in Federated Learning. In NeurIPS 2023.
Fedgh: Heterogeneous federated learning with generalized global header. In MM 2023.

**Questions:**

1. Would the authors explain how you chose the parameter $s$ in Section 5.6?

2. Why is the class $k$ missing in the proof in Appendix B (while formulated before proof)?

3. Can authors make more comparisons with recent (>=2023) works? For example,
Fedcp: Separating feature information for personalized federated learning via conditional policy. In KDD 2023.
FedFed: Feature Distillation against Data Heterogeneity in Federated Learning. In NeurIPS 2023.
Fedgh: Heterogeneous federated learning with generalized global header. In MM 2023.

It would be better to compare aggregation-based improvements in recent works and the new mechanism proposed in this paper. Since the authors stressed the drawbacks of these works, it would be better to give detailed support or explanation.

---

> ### Author Response · Authors · 2024-11-18
>
> Thank you for your insightful feedback and helpful suggestions, which have been invaluable in strengthening our manuscript. We have structured our response around the main issues identified, as detailed below.
>
> **W1: The introduction of differential privacy (DP) adds randomness to the convergence process of MPFT, requiring inclusion in the convergence proof.**
>
> **Response:**
> Refer to Theorem 1 and Appendix B of our paper. The only change after adding DP is for the term
> $$
>     \Delta_i^{(k)} = \|p_i^{(k)} - \bar{p}^{(k)}\|, \quad \bar{p}^{(k)} = \frac{1}{N} \sum_{i=1}^N p_i^{(k)},
> $$
>
> With $\Delta$ be the upper bound of all variance term $\Delta_i^{(k)}$. The Gaussian noise is added to each $p_i^{(k)}$. As the Gaussian distribution has mean zero, by law of large numbers we can assume $\bar{p}^{(k)}$ remains unchanged as $N$ is large. Suppose the noise term added to $p_i^{(k)}$ is  $\beta(p_i^{(k)})$, with upper bound $\beta$. The new variance term
> $$
> \tilde{\Delta}_i^{(k)} \leq \Delta_i^{(k)}+\beta
> $$
> $$
> \Delta \leq \Delta+\beta
> $$
> The new convergence formula adapted from Theorem 1 is
>
> $\mathcal{L}(\mathbb{D}^{\mathcal{P}}; \Theta^{\mathcal{G}}, A^{\mathcal{G}}_{t+1})$ $ \leq \mathcal{L}(\mathbb{D}^{\mathcal{P}}; \Theta^{\mathcal{G}}, A^{\mathcal{G}}_t)$
>
> $ - c \|\nabla_{A^{\mathcal{G}}} \mathcal{L}(\mathbb{D}^{\mathcal{P}}; \Theta^{\mathcal{G}}, A^{\mathcal{G}}_t)\|^2+ \frac{\Delta + \beta}{N},$
>
> This means that the constant term of convergence increase only linear with noise, which is acceptable. Meanwhile, the coefficient of term $\beta$ is $\frac{1}{N}$, which is negligible when $N$ is large, so the DP noise would not harm the convergence in theory.
>
> We add this part into the Appendix B.3 of our paper.
>
> **W2 and Q3: Include comparisons with recent works (≥2023), such as FedCP, FedFed, and FedGH.**
>
> **Response:** However, we implemented and compared FedCP and FedFed with MPFT on the subset50 dataset of DomainNet. The results are presented below:
>
> | Methods                  | ood acc | ind acc |
> |:------------------------:|:-------:|:-------:|
> | FedCP                    | 0.7755  | 0.718   |
> | FedFed                   | 0.6406  | 0.7817  |
> | MPFT (Average)           | 0.8077  | 0.7813  |
> | MPFT (Cluster, rate=0.1) | 0.7951  | 0.7957  |
> | MPFT (Cluster, rate=0.3) | 0.8204  | **0.8294**  |
> | MPFT (Random, rate=0.1)  | 0.7953  | 0.7899  |
> | MPFT (Random, rate=0.3)  | **0.8236**  | **0.8294**  |
>
> FedGH does not provide publicly available GitHub code, and its target scenario primarily addresses model heterogeneity, whereas MPFT focuses on scenarios involving heterogeneous client data domains. Therefore, we did not include FedGH in our experiments.

---

> > ### Author Response · Authors · 2024-11-18
> >
> > **Q1: Explain how to chose the parameter $s$ in Section 5.6 Differential Privacy.**
> >
> > **Response:** We perform differential privacy (DP) analysis in the average prototype sampling stage.
> > The following part are mainly adapted from Appendix A of DP textbook [1], numbering is same as in [2]:
> >
> > **Proposition 2.1 (Post-Processing).** Let $f : \mathbb{N}^{|\mathcal{X}|} \to R$ be a randomized algorithm that is $(\varepsilon, \delta)$-differentially private. Let $g : R \to R'$ be an arbitrary randomized mapping. Then $g \circ f : \mathbb{N}^{|\mathcal{X}|} \to R'$ is $(\varepsilon, \delta)$-differentially private.
> >
> > In our work, we can regard sampling as $f$ and learning from prototype as $g$ (the output of $g$ is the learned model), then the whole learning process is $(\varepsilon, \delta)$-differentially private provided that the sampling function $f$ is $(\varepsilon, \delta)$-differentially private.
> >
> >
> > **Definition A.1**  (Gaussian Mechanism) Let $f : \mathbb{N}^{|\mathcal{X}|} \to \mathbb{R}^d$ be an arbitrary $d$-dimensional function, and define its $\ell_2$ sensitivity to be:
> >
> > $$
> > \Delta_2 f = \max_{\text{adjacent } x, y} \|f(x) - f(y)\|_2.
> > $$
> >
> > The **Gaussian Mechanism** with parameter $\sigma$ adds noise scaled to $\mathcal{N}(0, \sigma^2)$ to each of the $d$ components of the output.
> >
> > **Theorem A.1**: Let $\varepsilon \in (0, 1)$ be arbitrary. For $c^2 > 2 \ln(1.25 / \delta)$, the Gaussian Mechanism with parameter
> >
> > $$
> > \sigma \geq c \frac{\Delta_2 f}{\varepsilon}
> > $$
> >
> > is $(\varepsilon, \delta)$-differentially private.
> >
> > For average sampling, the sampling procedure for class $i$, the input of function $f$ is data embeddings $D_i = \{d_k^{(i)}\}_{k=1}^{n_i}$, where $n_i$ is the number of data points in class $i$. For cluster sampling, just replace $D_i$ by data embeddings inside the cluster and other analysis unchanged.
> >
> > The average sampling function $f$ is
> > $$
> > f(D_i) = \frac{\sum_{k=1}^{n_i} d_k^{(i)}}{n_i}
> > $$
> > We define $f_i$ as the restriction of $f$ taking only datapoints in class $i$ as input. From [2], two adjacent datasets differs exactly one data point. For two adjacent datasets $D_p,D_q$, suppose they differ at datapoints $d_p^{(i)} \in D_p,d_q^{(i)} \in D_q$, then:
> > $$
> > \|f(D_p) - f(D_q)\|_2 = \frac{\left \| d_p^{(i)} - d_q^{(i)} \right \|_2}{n_i}
> > $$
> >
> > $$\triangle_2 f_i
> > =\max_{p,q \in\{1,...,n_i\}  } \frac{\left \| d_p^{(i)} - d_q^{(i)} \right \|_2}{n_i}$$
> >
> >
> > for $(\delta_i, \varepsilon_i)-DP$ guanrantee:
> >
> > $$
> > \sigma \ge \frac{\sqrt{2ln(1.25/\delta_i)} \cdot \triangle_2 f_i }{\varepsilon_i}
> > $$
> >
> > A strong DP gurantee is achieved when $\varepsilon_i<1$, $\delta_i \leq \frac{1}{n_i}$ according to the theorem in [2]. In federated learning settings such as [2], a medium DP gurantee is achieved when $\varepsilon_i$ is around 10.
> > We calculate our DP gurantee as
> > $$
> > \varepsilon_i = \frac{\sqrt{2ln(1.25n_i)} \cdot \triangle_2 f_i }{\sigma}
> > $$
> >
> > For our DP experiment, $\sigma$ value used is the multiplication $qs$ in Table 4 of our paper.
> > Here is our emperical results, we take the $\delta_i = \frac{1}{n_i}$ and report the average of $\varepsilon_i$:
> >
> > |            | $\sigma= 0.01$ | $\sigma= 0.05$ | $\sigma= 0.1$ | $\sigma= 0.5$ |
> > |:----------:|:--------------:|:--------------:|:-------------:|:-------------:|
> > | domainnet, subset=50  | 54.76          | 10.95          | 5.48          | 1.10          |
> > | domainnet, subset=100 | 65.48          | 13.10          | 6.55          | 1.31          |
> > | domainnet, subset=150 | 60.73          | 12.15          | 6.07          | 1.21          |
> >
> >
> > For our differential privacy (DP) experiment, the value of $\sigma$ used is the product $qs$, as described in Table 4 of our paper. Thus, we can determine $\sigma$ by first setting an ideal privacy budget $\varepsilon$ and then calculating the required noise $\sigma$ using the DP guarantee.
> >
> > We add this part into the Appendix H of our paper.
> >
> > **Q2: Address why the class $k$ is missing in the proof in Appendix B (while formulated before proof).**
> >
> > **Response:** As noted in the Appendix B's Assumptions section, we assume that $\Delta_i^{(k)} \leq \Delta$ for all $i$ and $k$. Based on this assumption, we use $\Delta$ as a universal bound to simplify the notation in the subsequent steps of the proof. This replacement ensures clarity and conciseness in the mathematical derivations. Since the prototypes are derived from same pretrained feature extractor, this assumption likely holds due to the consistency provided by the pretrained encoder in synchronizing the embedding space across domains.
> >
> > **Reference**
> >
> > [1] Dwork C, Roth A. The algorithmic foundations of differential privacy[J]. Foundations and Trends® in Theoretical Computer Science, 2014, 9(3–4): 211-407.
> >
> > [2] Wei K, Li J, Ding M, et al. Federated learning with differential privacy: Algorithms and performance analysis[J]. IEEE transactions on information forensics and security, 2020, 15: 3454-3469.

---

> ### Author Response · Authors · 2024-11-22
>
> Dear reviewer, we have submitted a rebuttal and would appreciate if you could check as the discussion phase is over soon. Thanks.

---

> > ### Comment · Reviewer_ckUs · 2024-11-27
> >
> > Thanks for the new revision in the paper and the response. My concerns have been addressed, and the score will be updated soon.

---

> > > ### Author Response · Authors · 2024-11-27
> > >
> > > I would like to sincerely thank you for the positive feedback and the score improvement (from 6 to 8). Your thoughtful comments and constructive suggestions were invaluable in strengthening our work, and I truly appreciate the time and effort you dedicated to reviewing our manuscript.

---

### Official Review · Reviewer_xVyS · 2024-10-31

**Soundness:** 3
**Presentation:** 3
**Contribution:** 1
**Rating:** 5
**Confidence:** 4

**Summary:**

The paper proposes Multi-domain Prototype-based Federated Fine-Tuning (MPFT), a framework designed for Federated Domain Adaptation (FDA) by utilizing domain-specific prototypes. Instead of relying on traditional model aggregation techniques, which often falter due to data heterogeneity, MPFT transfers prototypes (compressed domain representations) to the server. This approach allows the server to learn a global adapter that improves both in-domain and out-of-domain performance. MPFT also incorporates differential privacy to protect prototypes from potential data leakage.

**Strengths:**

The main strength of the paper lies in its comprehensive experimentation, which provides a detailed view of MPFT’s performance across various scenarios, showing superior computational and communication efficiency.

**Weaknesses:**

The Multi-domain Prototype-based Federated Fine-Tuning (MPFT) method proposed in this paper is overly simplistic, primarily relying on basic prototype sampling strategies—such as mean, cluster, and random sampling—to generate client prototypes. Additionally, because the model does not undergo training on each client’s data, it fundamentally contradicts the original design principles of federated learning (FL). The performance gains observed in the experiments likely stem from the strong pre-trained feature extractor, which does not require training on all client data but only a few representative prototypes to achieve reasonable performance. If the pre-trained feature extractor were removed and the model was trained solely on prototypes, the results would likely be poor. If only prototype fine-tuning is needed, then why even employ a federated learning framework? Thus, despite achieving certain results, this approach introduces no complex mechanisms or innovative architectures to enhance the FL system, lacking fresh or original design. It merely follows a standard process: feature extraction with a general pre-trained model, prototype selection, server training, and client fine-tuning, without presenting any real novelty.

The paper does not provide a thorough theoretical analysis of the different prototype sampling methods, nor does it uncover the fundamental differences between these sampling strategies in handling heterogeneous data distributions. Although the paper presents experimental results comparing mean, cluster, and random sampling methods, it lacks detailed explanations on the core differences among these strategies in terms of learning mechanisms, communication efficiency, and personalization effects. Consequently, the paper falls short in theoretical depth, failing to provide any deep insights into the impacts of these sampling strategies.

The theoretical analysis in this paper primarily focuses on convergence. However, training the adapter on aggregated prototype data is no different from centralized data training, so convergence is naturally expected. To provide valuable theoretical insights, the paper would need to show that the adapter’s performance on aggregated prototypes is comparable to that on aggregated raw client data or offer theoretical performance bounds. Furthermore, transmitting prototypes instead of models introduces a higher risk of privacy leakage, so the analysis should prioritize differential privacy (DP) guarantees rather than focusing solely on convergence. The current paper fails to address theoretical guarantees for privacy protection, which is particularly critical in privacy-sensitive federated learning contexts.

Although the experimental section is comprehensive, covering multiple datasets, various sampling strategies, and different DP configurations, it does not clearly reveal the functional differences and applicability of each prototype sampling approach across different scenarios. While the experiments test different prototype configurations, they lack an in-depth discussion on how these configurations perform under different data distribution patterns and sample sparsity conditions. This results in a set of experiments that, while extensive, fails to provide a robust summary of the unique characteristics of the prototype methods, thus missing an opportunity to elevate the research value.

**Questions:**

I suggest that the authors remove the overly powerful pre-trained feature extractor and instead use a simpler, more foundational model for feature extraction, such as ResNet or ConvNet. This would more accurately showcase the contribution of the prototype and methodology itself.

The authors should provide a validity proof to demonstrate that training the adapter on aggregated prototypes achieves comparable performance to training on aggregated raw client data or features—or, alternatively, establish an upper bound on the performance gap between these approaches.

Although differential privacy (DP) experiments were conducted, a theoretical analysis is also necessary. Specifically, the authors should clarify the noise variance conditions required for their method to satisfy DP. In particular, the statement “Furthermore, we observe that specific noise configurations can reduce bias across heterogeneous datasets, enhancing the robustness of prototype data” requires a more robust theoretical explanation.

I would encourage the authors to further explain, both theoretically and experimentally, the differences, advantages, and potential complementarities among various prototype selection mechanisms.

---

> ### Author Response · Authors · 2024-11-18
>
> We sincerely thank you for the insightful comments and valuable suggestions, which have greatly contributed to enhancing our manuscript. Based on the given weaknesses and questions, the key issues we address in our response include:
>
> **W1: It is necessary to explicitly clarify the innovations of the MPFT framework. As noted in the comment, "Thus, despite achieving certain results, this approach introduces no complex mechanisms or innovative architectures to enhance the FL system, lacking fresh or original design."**
>
> **Response:** Regarding the novelty of our paper, we highlight several key contributions of the MPFT framework that address important challenges in multi-domain federated learning scenarios:
>
> * **Prototype-based communication:** Instead of transmitting parameters or raw data, our framework transmits feature prototypes. This reduces communication costs while preserving privacy, as the prototypes are encoded representations of data rather than the raw data itself. Instead of transmitting parameters or raw data, our framework transmits feature prototypes, reducing communication costs while preserving privacy. The prototypes are encoded representations of data rather than raw data, ensuring secure and efficient communication. Unlike previous prototype-based methods in federated learning, such as FedProto, MPFT defines the prototype’s granularity at the class level, rather than at the client level as in FedProto. This approach effectively avoids the unreasonable aggregation of features from different classes into a single prototype, as discussed in Appendix A of the paper.
> * **Elimination of reliance on averaging-based aggregation:** While many existing FL frameworks depend on averaging-based aggregation of models or features, which assumes linear combinability of model parameters, our framework avoids this dependency. This makes MPFT particularly suitable for scenarios with high domain heterogeneity, where traditional averaging-based approaches often fail.
> * **Balancing generalization and personalization:** MPFT achieves a balance between global generalization and local personalization. The global model benefits from cross-client knowledge, while the local adaptation process (Algorithm 3) ensures high in-domain accuracy for individual clients. This is achieved with reduced communication overhead, making the framework more efficient.
>
> **W1: Using only a small number of prototypes to train the global model rather than utilizing each client’s local raw data may seem to contradict the foundational principles of federated learning (FL). Specifically, as noted, "Additionally, because the model does not undergo training on each client’s data, it fundamentally contradicts the original design principles of federated learning (FL)." and "If only prototype fine-tuning is needed, then why even employ a federated learning framework?"**
>
> **Response:** Federated learning (FL) is fundamentally designed with several key principles in mind. First, FL enables collaborative model training without the need to centralize raw data, thereby preserving data privacy by keeping data localized and transmitting only model updates or parameters [1][2][3]. This privacy-preserving feature is a core aspect of our approach, as MPFT does not transmit raw data during training. Instead, it transmits feature prototypes encoded and sampled by a pre-trained feature extractor, which are further protected through differential privacy. The use of prototypes as a replacement for raw data has been explored in prior work such as FedProto, demonstrating that leveraging prototypes aligns with the foundational principles of FL. Additionally, in Appendix G, we have designed potential attack scenarios to empirically demonstrate that this approach safeguards against the reconstruction of raw data details.
>
> Second, FL relies on global aggregation, allowing users to learn from each other’s representations. In our method, this principle is preserved through the use of prototypes as a means of sharing information between clients and the global model. This enables collaborative learning without compromising data privacy.
>
> Third, local adaptation is a commonly used feature in personalized FL, allowing for improved in-domain accuracy on local datasets [4]. Our approach includes a local adaptation process (Algorithm 3) that employs a knowledge distillation mechanism to reduce global knowledge forgetting while leveraging each client’s raw data locally. Therefore, contrary to the assertion that “the model does not undergo training on each client’s data,” our method does utilize local client's raw data for adaptation, ensuring that the global model remains personalized for each client.

---

> > ### Author Response · Authors · 2024-11-18
> >
> > **W1 and Q1: The use of a strong pre-trained feature extractor (such as the ViT series) in our method could lead to the impression that the effectiveness observed in experiments is largely due to the pre-training model rather than our approach and prototype contributions.**
> >
> > **Response:** To ensure a fair comparison, we clarify that all methods in the table, including FedAvg and other baseline methods, are evaluated using the same pre-trained feature extractor (e.g. ViT-B-32, ResNet50 or ConvNext). This setup allows for a direct and unbiased comparison across methods, addressing any concerns that our results may primarily reflect the benefits of a stronger pre-trained model.
> >
> > The following tables show that even when using weaker, more foundational pre-trained feature extractors (e.g. ResNet and ConvNext), our method (MPFT) consistently achieves the best performance in both out-of-domain (ood acc) and in-domain (ind acc) accuracy. While the overall accuracy is reduced due to the limitations of the weaker feature extractor, these results emphasize that the effectiveness of our approach is not dependent solely on the strength of the feature extractor, but rather on the contributions of our method itself.
> >
> > Table: Comparison of different FL methods Using ResNet50 as the Pre-trained Feature Extractor
> > | Methods                  | ood acc | ind acc |
> > |:------------------------:|:-------:|:-------:|
> > | local                    | 0.2069  | 0.6554  |
> > | Fedavg                   | 0.2535  | 0.4065  |
> > | FedProx                  | 0.2257  | 0.3597  |
> > | Ditto                    | 0.2241  | 0.3901  |
> > | MOON                     | 0.2601  | 0.4173  |
> > | Fedproto                 | 0.2105  | 0.4872  |
> > | DBE                      | 0.2587  | 0.4298  |
> > | MPFT (Average)           | 0.225   | 0.4367  |
> > | MPFT (Cluster, rate=0.1) | 0.245   | 0.5162  |
> > | MPFT (Cluster, rate=0.3) | **0.2672**  | 0.5484  |
> > | MPFT (Random, rate=0.1)  | 0.2329  | 0.4909  |
> > | MPFT (Random, rate=0.3)  | 0.2643  | **0.5528**  |
> >
> > Table: Comparison of different FL methods Using ConvNext as the Pre-trained Feature Extractor
> > | Methods                  | ood acc | ind acc |
> > |:------------------------:|:-------:|:-------:|
> > | local                    | 0.654  | 0.8696  |
> > | Fedavg                   | 0.7906  | 0.7274  |
> > | FedProx                  | 0.7766  | 0.7095  |
> > | Ditto                    | 0.779  | 0.7446  |
> > | MOON                     | 0.7914  | 0.7281  |
> > | Fedproto                 | 0.7154  | 0.802  |
> > | DBE                      | 0.754  | 0.7439  |
> > | MPFT (Average)           | 0.8071   | 0.7779  |
> > | MPFT (Cluster, rate=0.1) | 0.7823   | 0.7838  |
> > | MPFT (Cluster, rate=0.3) | **0.8144**  | **0.8145**  |
> > | MPFT (Random, rate=0.1)  | 0.7878  | 0.7766  |
> > | MPFT (Random, rate=0.3)  | 0.8132  | 0.809  |
> >
> > **W2, W4 and Q4: It is necessary to specify the exact differences between the sampling methods (average, cluster, and random) and to clarify the distinctions and potential complementarity among them.**
> >
> >
> > **Response:** We provide a detailed explanation of the differences, advantages, and suitable application scenarios for each sampling method:
> >
> > * **Average sampling** is particularly suitable for resource-constrained scenarios, as it transmits fewer prototypes. This leads to faster convergence and lower transmission costs during global adapter initialization (see Table 3 in the paper). It provides a straightforward and computationally efficient way to train a global model with reasonable generalization.
> > * **Random sampling** works best when client domains are relatively independent, meaning domain heterogeneity across clients is significant. Random sampling captures more edge prototypes, which enhances the robustness of the sampling. However, in mixed domain ratio scenarios, random sampling may inadvertently undersample certain domains, potentially impacting its performance. Despite this, Section 5.1 of the paper demonstrates that random sampling performs well in scenarios with high cross-client domain heterogeneity.
> > * **Cluster sampling** automatically separates data domains with significant internal differences within the same client. It is particularly effective in scenarios with low client domain heterogeneity, where client domains are more blended. Section 5.2 of the paper empirically validates that cluster sampling achieves the best results in these mixed client domain scenarios (i.e., scenarios where one client has multiple sources of data domains).

---

> > > ### Author Response · Authors · 2024-11-18
> > >
> > > **W3 and Q2: In both theoretical and experimental aspects, we need to clarify the differences between training on original data and training on a subset of prototypes.**
> > >
> > > **Response:** The use of prototypes can be theoretically supported by drawing parallels to established machine learning techniques. In coreset selection, a small representative subset of data is chosen to approximate the performance of the entire dataset, reducing computational costs while maintaining model quality. Similarly, in our framework, prototypes serve as a representative subset of the original data distribution. Support Vector Machines (SVMs) rely on selecting a subset of data points (support vectors) that are critical for defining the decision boundary, which aligns with our approach, where prototypes act as key representatives to capture essential features of the data. Additionally, in Stochastic Gradient Descent (SGD), gradients are computed using a mini-batch of data rather than the entire dataset, approximating the overall data distribution. Similarly, our method uses prototypes to approximate the behavior of the full dataset. These align with the principle that a carefully selected subset can approximate the behavior of the entire dataset.
> > >
> > > To empirically validate this, we varied the sample rate of prototypes (i.e., the proportion of data retained as prototypes) and compared the performance with centralized learning (using the entire dataset) in the following table. The results demonstrate that as the sample rate increases, both out-of-domain (OOD) and in-domain (IND) accuracy improve and closely approximate the performance of centralized learning. Even with lower sample rates, the accuracy remains competitive, highlighting that prototypes effectively capture critical information needed for training while reducing communication costs and preserving privacy.
> > >
> > > Table: Performance Comparison of Prototype-Based Training with Different Sample Rates and Centralized Learning
> > > | Sample Rate | Sample Method | OOD Accuracy | IND Accuracy | Sample Method | OOD Accuracy | IND Accuracy |
> > > |------------|---------------|--------------|--------------|---------------|--------------|--------------|
> > > | 0.1        | cluster       | 0.7951       | 0.7957       | random        | 0.7953       | 0.7899       |
> > > | 0.3        | cluster       | 0.8204       | 0.8294       | random        | 0.8236       | 0.8294       |
> > > | 0.5        | cluster       | 0.8256       | 0.8289       | random        | 0.8294       | 0.8341       |
> > > | 0.7        | cluster       | 0.8339       | 0.8386       | random        | 0.8387       | 0.8442       |
> > > | centralized learning |  /  | 0.8405       | 0.8432       | /             | 0.8405       | 0.8432       |

---

> > > > ### Author Response · Authors · 2024-11-18
> > > >
> > > > **W3 and Q3: It is necessary to analyze Differential Privacy (DP) from a theoretical perspective, providing a more robust theoretical explanation rather than relying solely on experimental comparisons.**
> > > >
> > > > **Response:**
> > > > We perform differential privacy (DP) analysis in the average prototype sampling stage.
> > > > The following part are mainly adapted from Appendix A of DP textbook [5], numbering is same as in [5]:
> > > >
> > > > **Proposition 2.1 (Post-Processing).** Let $f : \mathbb{N}^{|\mathcal{X}|} \to R$ be a randomized algorithm that is $(\varepsilon, \delta)$-differentially private. Let $g : R \to R'$ be an arbitrary randomized mapping. Then $g \circ f : \mathbb{N}^{|\mathcal{X}|} \to R'$ is $(\varepsilon, \delta)$-differentially private.
> > > >
> > > > In our work, we can regard sampling as $f$ and learning from prototype as $g$ (the output of $g$ is the learned model), then the whole learning process is $(\varepsilon, \delta)$-differentially private provided that the sampling function $f$ is $(\varepsilon, \delta)$-differentially private.
> > > >
> > > >
> > > > **Definition A.1**  (Gaussian Mechanism) Let $f : \mathbb{N}^{|\mathcal{X}|} \to \mathbb{R}^d$ be an arbitrary $d$-dimensional function, and define its $\ell_2$ sensitivity to be:
> > > >
> > > > $$
> > > > \Delta_2 f = \max_{\text{adjacent } x, y} \|f(x) - f(y)\|_2.
> > > > $$
> > > >
> > > > The **Gaussian Mechanism** with parameter $\sigma$ adds noise scaled to $\mathcal{N}(0, \sigma^2)$ to each of the $d$ components of the output.
> > > >
> > > > **Theorem A.1**: Let $\varepsilon \in (0, 1)$ be arbitrary. For $c^2 > 2 \ln(1.25 / \delta)$, the Gaussian Mechanism with parameter
> > > >
> > > > $$
> > > > \sigma \geq c \frac{\Delta_2 f}{\varepsilon}
> > > > $$
> > > >
> > > > is $(\varepsilon, \delta)$-differentially private.
> > > >
> > > > For average sampling, the sampling procedure for class $i$, the input of function $f$ is data embeddings $D_i = \{d_k^{(i)}\}_{k=1}^{n_i}$, where $n_i$ is the number of data points in class $i$. For cluster sampling, just replace $D_i$ by data embeddings inside the cluster and other analysis unchanged.
> > > >
> > > > The average sampling function $f$ is
> > > > $$
> > > > f(D_i) = \frac{\sum_{k=1}^{n_i} d_k^{(i)}}{n_i}
> > > > $$
> > > > We define $f_i$ as the restriction of $f$ taking only datapoints in class $i$ as input. From [6], two adjacent datasets differs exactly one data point. For two adjacent datasets $D_p,D_q$, suppose they differ at datapoints $d_p^{(i)} \in D_p,d_q^{(i)} \in D_q$, then:
> > > > $$
> > > > \|f(D_p) - f(D_q)\|_2 = \frac{\left \| d_p^{(i)} - d_q^{(i)} \right \|_2}{n_i}
> > > > $$
> > > >
> > > > $$\triangle_2 f_i
> > > > =\max_{p,q \in\{1,...,n_i\}  } \frac{\left \| d_p^{(i)} - d_q^{(i)} \right \|_2}{n_i}$$
> > > >
> > > >
> > > > for $(\delta_i, \varepsilon_i)-DP$ guanrantee:
> > > >
> > > > $$
> > > > \sigma \ge \frac{\sqrt{2ln(1.25/\delta_i)} \cdot \triangle_2 f_i }{\varepsilon_i}
> > > > $$
> > > >
> > > > A strong DP gurantee is achieved when $\varepsilon_i<1$, $\delta_i \leq \frac{1}{n_i}$ according to the theorem in [6]. In federated learning settings such as [6], a medium DP gurantee is achieved when $\varepsilon_i$ is around 10.
> > > > We calculate our DP gurantee as
> > > > $$
> > > > \varepsilon_i = \frac{\sqrt{2ln(1.25n_i)} \cdot \triangle_2 f_i }{\sigma}
> > > > $$
> > > >
> > > > For our DP experiment, $\sigma$ value used is the multiplication $qs$ in Table 4 of our paper.
> > > > Here is our emperical results, we take the $\delta_i = \frac{1}{n_i}$ and report the average of $\varepsilon_i$:
> > > >
> > > > |            | $\sigma= 0.01$ | $\sigma= 0.05$ | $\sigma= 0.1$ | $\sigma= 0.5$ |
> > > > |:----------:|:--------------:|:--------------:|:-------------:|:-------------:|
> > > > | domainnet, subset=50  | 54.76          | 10.95          | 5.48          | 1.10          |
> > > > | domainnet, subset=100 | 65.48          | 13.10          | 6.55          | 1.31          |
> > > > | domainnet, subset=150 | 60.73          | 12.15          | 6.07          | 1.21          |
> > > >
> > > > The table above demonstrates that, as the noise scale $\sigma$ increases, the average privacy budget $\bar{\varepsilon}$ decreases, indicating stronger privacy protection. When $\sigma$ is above 0.05, the privacy budget is sufficient to provide robust privacy guarantees. Furthermore, as shown in Table 4 of our paper, the performance of MPFT experiences minimal decline when $\sigma$ is in the range of 0.05–0.1. These results, both theoretical and empirical, demonstrate that MPFT can effectively utilize differential privacy mechanisms to protect prototypes without significant performance loss.
> > > >
> > > > We add this part into the Appendix H of our paper.

---

> > > > > ### Author Response · Authors · 2024-11-18
> > > > >
> > > > > **Reference**
> > > > >
> > > > > [1] McMahan, H. B., Moore, E., Ramage, D., Hampson, S., & Arcas, B. A. y. (2017). Communication-Efficient Learning of Deep Networks from Decentralized Data. In Artificial Intelligence and Statistics (pp. 1273–1282). PMLR.
> > > > >
> > > > > [2] IBM Research. What is Federated Learning? https://research.ibm.com/blog/what-is-federated-learning
> > > > >
> > > > > [3] NVIDIA. What is Federated Learning? https://blogs.nvidia.com/blog/what-is-federated-learning/
> > > > >
> > > > > [4] Li, T., Sanjabi, M., Beirami, A., & Smith, V. (2020). Salvaging Federated Learning by Local Adaptation. In Proceedings of the 37th International Conference on Machine Learning (pp. 10157–10166).
> > > > >
> > > > > [5] Dwork C, Roth A. The algorithmic foundations of differential privacy[J]. Foundations and Trends® in Theoretical Computer Science, 2014, 9(3–4): 211-407.
> > > > >
> > > > > [6] Wei K, Li J, Ding M, et al. Federated learning with differential privacy: Algorithms and performance analysis[J]. IEEE transactions on information forensics and security, 2020, 15: 3454-3469.

---

> ### Author Response · Authors · 2024-11-22
>
> Dear reviewer, we have submitted a rebuttal and would appreciate if you could check as the discussion phase is over soon. Thanks.

---

### Official Review · Reviewer_ZWk6 · 2024-11-04

**Soundness:** 3
**Presentation:** 3
**Contribution:** 2
**Rating:** 6
**Confidence:** 3

**Summary:**

This work studies domain adaptation in federated learning scenarios, employing prototype-based fine-tuning to leverage knowledge from other clients. The fine-tuning procedure requires only one round of communication between clients and the server, making it communication-computationally efficient.

**Strengths:**

1, Experiments on two datasets indicate a significant performance boost from the proposal, increasing accuracy by 2%.

2. The approach is intuitive, sharing prototypes using three sampling strategies.

3. The proposal is communication-computationally efficient, as it involves a low number of interaction rounds and requires only prototypes for communication.

**Weaknesses:**

The motivating scenario may be impractical, as the proposal assumes that clients already possess a well-pretrained model. This assumption can be particularly challenging in sensitive areas where federated learning is essential, such as healthcare or finance, where data privacy concerns limit access to robust pre-trained models.

Can the authors discuss specific applications or scenarios within these sensitive domains where obtaining a pre-trained model might be more feasible? Additionally, it would be valuable to explore potential solutions or adaptations of their method for situations where a pre-trained model is not available.

**Questions:**

1. The experimental settings are somewhat vague, particularly regarding whether competitive approaches, such as FedAvg and FedProx, are trained from scratch or from the same pretrained backbone model as the MPFT. Could the authors specify this information in the paper, ideally in the experimental setup or implementation details section?

2. The privacy protection measures appear informal. While Gaussian noise is added to prototypes (i.e., embeddings) as noted in Section 5.6, it is important to assess whether the noise level provides adequate privacy protection for sensitive attributes or labels. To strengthen this section, could the authors provide a more rigorous analysis of the privacy guarantees associated with their approach? For example, quantifying the level of privacy protection using established metrics or frameworks would be beneficial.

---

> ### Author Response · Authors · 2024-11-18
>
> We greatly appreciate your thoughtful comments and constructive feedback, which have been invaluable in improving our manuscript. To address the identified weaknesses and questions, we have focused our responses on the following key issues:
>
> **W1: Discuss alternative solutions for scenarios in sensitive domains (such as healthcare and finance) where obtaining a well-pretrained model is challenging due to privacy concerns.**
>
> **Response:** We acknowledge the concern regarding difficulty in acquire pretrained models in sensitive domains, such as healthcare and finance, due to privacy restrictions. However, the models used in our experiments, such as ViT, are open-source and widely accessible. Moreover, in response to reviewer xVyS's comment **W1 and Q1**, we use a weaker and more widely-used pretrained feature extractor ResNet-50 and still get the best results among all baselines.
>
> Additionally, pretrained models are increasingly used in sensitive areas. In healthcare, models like BioBERT [1] support medical tasks such as biomedical text mining. While in finance, models like FinBERT [2] assist with sentiment analysis of financial texts and other nlp tasks in the financial domain.
>
> These illustrate that pretrained models are not only accessible but also extensively adopted in sensitive domains, where their ability to leverage prior knowledge from publicly available datasets provides significant advantages. Therefore, we believe that our use of pretrained models is both practical and aligned with current trends in these domains.
>
> **Q1: Clarify the experimental setup and implementation, particularly whether baselines like FedAvg and FedProx are trained from scratch or use the same pre-trained backbone as MPFT.**
>
> **Response:** To ensure fairness in the comparison, all baseline methods, including FedAvg and FedProx, were trained using the same pre-trained feature extractor as MPFT. Furthermore, the training setup for all baselines was aligned with MPFT’s training paradigm: the pre-trained feature extractor and the global head were frozen, and only the adapter was trained.
>
> For consistency, operations in other baselines (e.g., average aggregation in FedAvg) were applied specifically to the adapter parameters instead of the entire model parameters. This adaptation ensures a fair comparison of the performance of MPFT and the baselines under the same conditions. We have clarified these details in the revised manuscript, adding the experimental setup to Appendix C.2.

---

> > ### Author Response · Authors · 2024-11-18
> >
> > **Q2: Provide a more robust analysis of the differential privacy section by quantifying the relationship between added noise and the privacy budget.**
> >
> > **Response:**
> > We perform differential privacy (DP) analysis in the average prototype sampling stage.
> > The following part are mainly adapted from Appendix A of DP textbook [3], numbering is same as in [3]:
> >
> > **Proposition 2.1 (Post-Processing).** Let $f : \mathbb{N}^{|\mathcal{X}|} \to R$ be a randomized algorithm that is $(\varepsilon, \delta)$-differentially private. Let $g : R \to R'$ be an arbitrary randomized mapping. Then $g \circ f : \mathbb{N}^{|\mathcal{X}|} \to R'$ is $(\varepsilon, \delta)$-differentially private.
> >
> > In our work, we can regard sampling as $f$ and learning from prototype as $g$ (the output of $g$ is the learned model), then the whole learning process is $(\varepsilon, \delta)$-differentially private provided that the sampling function $f$ is $(\varepsilon, \delta)$-differentially private.
> >
> >
> > **Definition A.1**  (Gaussian Mechanism) Let $f : \mathbb{N}^{|\mathcal{X}|} \to \mathbb{R}^d$ be an arbitrary $d$-dimensional function, and define its $\ell_2$ sensitivity to be:
> >
> > $
> > \Delta_2 f = \max_{\text{adjacent } x, y} \|f(x) - f(y)\|_2.
> > $
> >
> > The **Gaussian Mechanism** with parameter $\sigma$ adds noise scaled to $\mathcal{N}(0, \sigma^2)$ to each of the $d$ components of the output.
> >
> > **Theorem A.1**: Let $\varepsilon \in (0, 1)$ be arbitrary. For $c^2 > 2 \ln(1.25 / \delta)$, the Gaussian Mechanism with parameter
> >
> > $$
> > \sigma \geq c \frac{\Delta_2 f}{\varepsilon}
> > $$
> >
> > is $(\varepsilon, \delta)$-differentially private.
> >
> > For average sampling, the sampling procedure for class $i$, the input of function $f$ is data embeddings $D_i = \{d_k^{(i)}\}_{k=1}^{n_i}$, where $n_i$ is the number of data points in class $i$. For cluster sampling, just replace $D_i$ by data embeddings inside the cluster and other analysis unchanged.
> >
> > The average sampling function $f$ is
> > $$
> > f(D_i) = \frac{\sum_{k=1}^{n_i} d_k^{(i)}}{n_i}
> > $$
> > We define $f_i$ as the restriction of $f$ taking only datapoints in class $i$ as input. From [4], two adjacent datasets differs exactly one data point. For two adjacent datasets $D_p,D_q$, suppose they differ at datapoints $d_p^{(i)} \in D_p,d_q^{(i)} \in D_q$, then:
> >
> > $$
> > \|f(D_p) - f(D_q)\|_2 = \frac{\left | d_p^{(i)} - d_q^{(i)} \right |_2}{n_i}
> > $$
> >
> >
> > $$\triangle_2 f_i
> > =\max_{p,q \in\{1,...,n_i\}  } \frac{\left \| d_p^{(i)} - d_q^{(i)} \right \|_2}{n_i}$$
> >
> >
> > for $(\delta_i, \varepsilon_i)-DP$ guanrantee:
> >
> > $$
> > \sigma \ge \frac{\sqrt{2ln(1.25/\delta_i)} \cdot \triangle_2 f_i }{\varepsilon_i}
> > $$
> >
> > A strong DP gurantee is achieved when $\varepsilon_i<1$, $\delta_i \leq \frac{1}{n_i}$ according to the theorem in [4]. In federated learning settings such as [4], a medium DP gurantee is achieved when $\varepsilon_i$ is around 10.
> > We calculate our DP gurantee as
> > $$
> > \varepsilon_i = \frac{\sqrt{2ln(1.25n_i)} \cdot \triangle_2 f_i }{\sigma}
> > $$
> >
> > For our DP experiment, $\sigma$ value used is the multiplication $qs$ in Table 4 of our paper.
> > Here is our emperical results, we take the $\delta_i = \frac{1}{n_i}$ and report the average of $\varepsilon_i$:
> >
> > |            | $\sigma= 0.01$ | $\sigma= 0.05$ | $\sigma= 0.1$ | $\sigma= 0.5$ |
> > |:----------:|:--------------:|:--------------:|:-------------:|:-------------:|
> > | domainnet, subset=50  | 54.76          | 10.95          | 5.48          | 1.10          |
> > | domainnet, subset=100 | 65.48          | 13.10          | 6.55          | 1.31          |
> > | domainnet, subset=150 | 60.73          | 12.15          | 6.07          | 1.21          |
> >
> > The table above demonstrates that, as the noise scale $\sigma$ increases, the average privacy budget $\bar{\varepsilon}$ decreases, indicating stronger privacy protection. When $\sigma$ is above 0.05, the privacy budget is sufficient to provide robust privacy guarantees. Furthermore, as shown in Table 4 of our paper, the performance of MPFT experiences minimal decline when $\sigma$ is in the range of 0.05–0.1. These results, both theoretical and empirical, demonstrate that MPFT can effectively utilize differential privacy mechanisms to protect prototypes without significant performance loss.
> >
> > We add this part into the Appendix H of our paper.

---

> > > ### Author Response · Authors · 2024-11-18
> > >
> > > **Reference**
> > >
> > > [1] Jinhyuk Lee, Wonjin Yoon, Sungdong Kim, Donghyeon Kim, Sunkyu Kim, Chan Ho So, and Jaewoo Kang. "BioBERT: a pre-trained biomedical language representation model for biomedical text mining." Bioinformatics 36, no. 4 (2020): 1234-1240.
> > >
> > > [2] Liu, Zhuang, et al. "Finbert: A pre-trained financial language representation model for financial text mining." Proceedings of the twenty-ninth international conference on international joint conferences on artificial intelligence. 2021.
> > >
> > > [3] Dwork C, Roth A. The algorithmic foundations of differential privacy[J]. Foundations and Trends® in Theoretical Computer Science, 2014, 9(3–4): 211-407.
> > >
> > > [4] Wei K, Li J, Ding M, et al. Federated learning with differential privacy: Algorithms and performance analysis[J]. IEEE transactions on information forensics and security, 2020, 15: 3454-3469.

---

> ### Author Response · Authors · 2024-11-22
>
> Dear reviewer, we have submitted a rebuttal and would appreciate if you could check as the discussion phase is over soon. Thanks.

---

### Author Response · Authors · 2024-11-24

Thank you for taking the time to review our manuscript and provide your valuable feedback. We have carefully addressed all the comments and incorporated changes to improve the manuscript accordingly. Below, we summarized responses to the most frequently raised concerns:

1. **Applicability of Pretrained Models (Reviewer ZWk6 - Weakness, Reviewer xVyS - W1, Q1):**
   - We clarified the feasibility of using pretrained models in sensitive domains like healthcare (e.g., BioBERT) and finance (e.g., FinBERT).
   - Additional experiments with weaker pretrained models (e.g., ResNet-50) validated that our method's effectiveness does not solely rely on strong feature extractors. The results, included in Section 5.1 and Appendix C.3, showed that MPFT outperforms baselines regardless of the feature extractor's strength.

2. **Experimental Details and Fair Comparisons (Reviewer ZWk6 - Q1, Reviewer xVyS - Q1):**
   - Clarified that all baseline methods, including FedAvg and FedProx, were trained with the same pretrained backbone as MPFT.
   - We explicitly outlined experimental setups and parameter configurations in Appendix C.2 of the revised manuscript.

3. **More Detailed Differential Privacy (DP) Analysis (Reviewer ZWk6 - Q2, Reviewer xVyS - W3, Q3, Reviewer ckUs - W1):**
   - We conducted a detailed theoretical analysis linking Gaussian noise to privacy budgets and empirically validated privacy guarantees.
   - Results in Appendix H confirm that our DP mechanism effectively protects prototypes while maintaining performance, even with increased noise levels.

4. **Novelty of MPFT Framework (Reviewer xVyS - W1):**
     - MPFT transmits feature prototypes instead of raw data or parameters, reducing communication costs and preserving privacy. MPFT uses class-level prototypes, avoiding the aggregation of features from different classes into a single prototype (Appendix A).
     - MPFT eliminates reliance on traditional averaging-based methods, making it more suitable for scenarios with high domain heterogeneity.
     - MPFT balances global generalization through aggregated prototypes and local adaptation via Algorithm 3, ensuring high in-domain accuracy with minimal communication overhead.

5. **More Detailed Comparison of Sampling Methods (Reviewer xVyS - W2, Q4):**
   - We analyzed the differences among average, random, and cluster sampling.
   - Cluster sampling performed best in mixed-domain settings, while random sampling excelled with high domain heterogeneity. These insights, supported by Table 3, are added to Section 5.3.

6. **Comparisons with More Recent Works (Reviewer ckUs - W2, Q3):**
   - We implemented FedCP and FedFed, showing that MPFT outperforms these methods on DomainNet subset50 (see Table 5).
   - FedGH was excluded due to unavailable code and different research focus (model heterogeneity). Details are clarified in Appendix G.

We deeply appreciate your efforts and would be happy to further clarify or discuss any remaining concerns. Please feel free to let us know if additional modifications or explanations are required.

---

### Meta-Review · Area_Chair_kEgr · 2024-12-15

**Metareview:**

This paper proposes to enhance federated domain adaptation and, in particular, to mitigate the challenge of data heterogeneity. The authors propose a framework known as Multi-domain Prototype-based Federated Fine-Tuning (MPFT). It finetunes a pre-trained models using multidomain prototypes. MPFT enables supervised learning on the server to derive a globally optimized adapter that is subsequently distributed to local clients, without the intrusion of data privacy. It is shown empirically that MPFT improves in- and out-of-domain accuracy over conventional federated domain adaptation methods. Privacy is also considered.

Generally, the reviewers felt positive about the paper, which uses some logical and natural approaches to solve the problem. While nothing is overly surprising, the theoretical and experimental results are sound. MPFT is low cost, efficient and robust. I believe that MPFT may be useful in future federated domain adaptation tasks.

**Additional Comments On Reviewer Discussion:**

The authors prepared detailed rebuttals to the reviewers, which are greatly appreciated. The authors also performed several additional experiments and theoretical analyses. Reviewer ckUs raised their score as a result. For the other responses, we did not hear back from the reviewers, but I assessed the responses and they add value to the paper. I encourage the authors to incorporate these into the final version of the paper.

---

### Decision · Program_Chairs · 2025-01-22

Accept (Poster)